

# An Approach to Track Instrument Calibration and Produce Consistent Products with the Version-8 Total Column Ozone Algorithm (V8TOZ)

Zhihua Zhang[1], Jianguo Niu[1], Lawrence E Flynn[2], Eric Beach[1], Trevor Beck[2]

[1]I. M. System Group Inc. at NOAA, College Park, MD 20740 USA

[2]Center for Satellite Application and Research, NOAA/NESDIS, College Park, MD 20740 USA

*Correspondence to*: Zhihua Zhang (Zhihua.zhang@noaa.gov)

**Abstract.** The Ozone Mapping and Profiler Suite (OMPS) has been onboard the Suomi National Polar-orbiting Partnership (S-NPP) satellite since October 2011, and was followed by an OMPS on NOAA-20 (N20) in November 2017, as part of the US Joint Polar Satellite System (JPSS) program. The OMPS measurements are processed to yield various products of

atmospheric composition data for near-real-time monitoring and off-line study, including retrievals of total column ozone (TCO) and an Ultraviolet absorbing aerosol index (AI) based on the version-8 total ozone (V8TOZ) algorithm. With the implementation of changes to employ a broadband channel approach in the NOAA OMPS V8TOZ, the retrieved TCO and AI products become more stable and consistent between S-NPP and N20. Two particular regions have been chosen for building soft-calibration adjustments for both OMPS S-NPP and N20, which force the V8TOZ retrievals to be in quite good agreement

from both sensors with little change by seasons. However, bias analysis shows that some noticeable errors / differences still exist after soft-calibration, and those errors appear to be quite persistently associated with solar zenith angle (SZA) and satellite viewing angle (SVA) in the retrievals of TCO and AI for both OMPS S-NPP and N20. Comparisons of TCO and AI from NOAA OMPS retrievals with other products such as those from the Tropospheric Monitoring Instrument (TROPOMI) and the Earth Polychromatic Imaging Camera (EPIC), show that, although the sensor, algorithm and solar spectra are different among

them, the overall retrievals from those products are quite similar and consistent.

## 1 Introduction

Ozone and aerosol loading in the atmosphere play an important role in environment and climate change, which require a broad set of actions across the world for monitoring and assessing their impacts. Observations from ground-based instruments can regularly provide continuous time series data, but they are spatially scattered with limited global coverage. In contrast, satellite

instruments have an important advantage for ozone and aerosol measurements, they can provide daily global ozone and aerosol maps with a resolution that is sufficient to detect meteorological variability across regions.

   Global-scale satellite observations of total column ozone (TCO) have been performed since the early 1970s, and regular and continuous ozone monitoring by the Total Ozone Mapping Spectrometer (TOMS) and Solar Backscatter Ultraviolet (SBUV) instruments onboard the Nimbus- 7 satellite started in 1978 (McPeters et al., 1996; Bhartia et al., 2013) . Since then, instruments

for ozone observations have been available on various platforms. Some of these instruments are the Global Ozone Monitoring



Experiment (GOME) onboard the European Remote Satellite-2 (ERS-2) (Bodeker et al., 2001), the Ozone Monitoring Instrument (OMI) onboard the Earth Observation System Aura satellite (Koukouli et al., 2012), the Earth Polychromatic Imaging Camera (EPIC) onboard the NOAA Deep Space Climate Observatory (DSCOVR) spacecraft (Marshak et al., 2018, Kramarova et al., 2021), and the TROPOspheric Monitoring Instrument (TROPOMI) onboard the Sentinel-5 Precursor (S5P)

mission (Lindfors et al., 2018, Inness et al., 2019, Garane et al., 2019). The concept of the UV absorbing aerosol index (AI) was initially introduced in the context of observations made by TOMS in the late 1990s for the correction of aerosol induced errors in the retrieval of total ozone (Herman et al., 1997; Torres and Bhartia, 1999). Since then it has been extended to apply to measurements with OMI (Herman et al., 1997; Torres et al., 1998, 2007). Using AI for detecting aerosol has been applied to other sensors such as GOME (de Graaf et al., 2005), EPIC (Lyapustin et al., 2021) and TROPOMI (Lindfors et al., 2018,

Kooreman et al., 2020). The channel wavelengths selected for deriving AI may differ from different sensors but the method and the purpose of generating AI in the ozone retrieval algorithms remain similar.

As one of the instruments in the US Joint Polar Satellite System (JPSS) program, the Ozone Mapping and Profiler Suite (OMPS) Nadir Mapper (OMPS-NM) was designed for total column ozone (TCO) and aerosol index (AI) retrievals. The first OMPS has been onboard the Suomi National Polar-orbiting Partnership (S-NPP) spacecraft since October 2011, the second

OMPS is flying on NOAA-20 (N20) launched in November 2017, and a third OMPS is flying on NOAA-21 launched in November 2022. All three platforms have orbital adjustments to maintain their 13:30 Equator crossing times. The OMPS-NM is a total ozone column sensor and uses a single grating and a charge-coupled device (CCD) array detector to make measurements every 0.42nm from 300 to 380nm with 1.0nm full width at half maximum (FWHM) resolution. It has a 110° cross-track field of view (FOV) (~2800 km on the Earth's surface) and 0.27° along-track slit width FOV. In standard Earth

science mode, the measurements are combined into 35 cross-track bins [20 spatial pixels giving 3.35° (50 km) at nadir, and 2.84° at ±55° cross-track dimensions for the FOVs]. The resolution for the OMPS-NM is changeable. For OMPS S-NPP, the resolution is 50x50 km along track at nadir, created by using a 7.6 s reporting/integration period. While for OMPS N20, the resolution is 50x17 km along track at nadir, created by using a 2.5 s integration period (Flynn et al., 2014; Seftor et al., 2014; Jaross et al., 2014; Wu et al., 2014; Pan et al., 2019).

The S-NPP OMPS-NM was reprocessed with a consistent set of calibration to produce a homogeneous SDR data set (Zou et al., 2020; Yan et al., 2022). The OMPS-NM instrument stability is monitored by using a pair of solar diffusers – a working diffuser used every two weeks and a reference diffuser used once per year. Taken together, the measurements track both the per-exposure diffuser degradation and the instrument throughput degradation. The latter is shared with the Earth radiance measurements. Analysis of the ten-year solar measurement record reveals that the S-NPP OMPS-NM radiometric calibration

has been stable at better than 1% as shown in Figure 1.



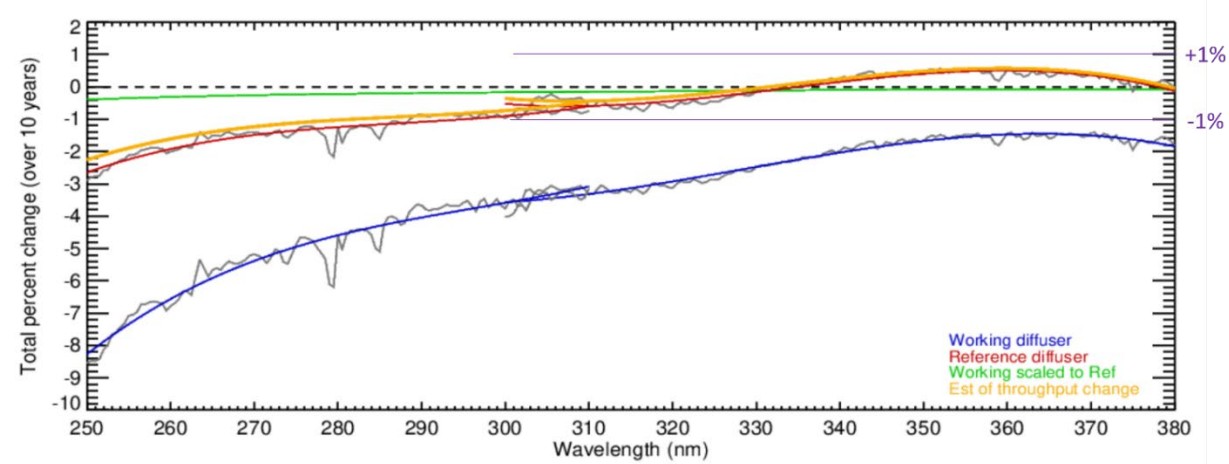

**Figure 1.** Estimates of the total wavelength-dependent throughput changes for the S-NPP OMPS-NP and OMPS-NM over ten years (2012 to 2022). The blue curve is from linear fits of the changes of the bi-weekly solar measurements from the working diffuser. The red curve is from linear fits of the changes of the annual solar measurements from the reference diffuser. The green curve is a scaling of the blue curve accounting for the difference in exposure frequency for the reference versus the working diffusers. The orange curve is the red curve minus the green curve. It gives an estimate of the throughput degradation for the shared optical path for the radiances measurements. Notice that the instrument throughput changes for the OMPS-NM (300 nm to 380 nm) are well within the ±1% level. (This figure was created and provided by Colin Seftor of SSAI for the NASA GSFC Ozone Team.)

To maintain consistency and continuity of retrieved ozone and aerosol index for climate data and atmospheric model studies, we employed a NASA developed Version 8 ozone retrieval algorithm (V8TOZ) (Wellemeyer et al., 1997; Bhartia & Wellemeyer, 2002; McPeters et al., 1996; Bhartia et al., 2013), for NOAA operational OMPS S-NPP and N20 retrievals as well as for off-line studies. Undergoing three decades of progressive refinement, V8TOZ has been used as the primary algorithm in the previous NOAA series of SBUV/2 products, and is now widely used for ozone retrievals and studies for many satellites. The science basis and statistical procedures as well as error sources for the V8 algorithm have been well documented in the OMPS ATBD and other articles (Bhartia & Wellemeyer, 2002; McPeters et al., 1996). Thanks to the OMPS series, which provide similar instruments with the same scanning method and the same local Equator crossing times with fixed measurement departure, the retrievals from OMPS S-NPP V8TOZ and N20 V8TOZ can be used for further analysis of biases. Those biases exist in the algorithm with various sources and are difficult to remove by soft-calibration. We provide a quantification of those differences with latitude. Researchers interested in error analysis and refined retrievals could take it as reference. This paper is structured as follows. Section 2 describes the retrieval algorithm and the differences between using broadband and narrowband approaches. Section 3 shows procedures of generating soft-calibration adjustments for both OMPS S-NPP and N20 to make their retrievals in agreement with each other, and exhibits stability and consistency of those two products through verification. Section 4 describes the potential biases that remain in the retrievals after soft-calibration adjustment. Section 5 shows comparison of OMPS retrieved total column ozone and aerosol index with the products from TROPOMI and EPIC. Section 6 gives the summary and conclusions.



## 2 V8TOZ with a broader bandpass approach

Based on the nature of the backscatter ultraviolet (BUV) radiance, two key assumptions are employed in the V8TOZ algorithm (Klenk et al., 1982; McPeters et al., 1996; Wellemeyer et al., 1997). The first assumption is that the BUV radiances at wavelengths greater than 310 nm are primarily a function of total ozone amount, with only a weak dependence on ozone profile
shapes that can be accounted for by using a set of climatological profiles. The second assumption is that a relatively simple radiative transfer model that treats clouds, aerosols, and surfaces as Lambertian reflectors can account for most of the spectral dependence of BUV radiation. Unlike the Version 8 ozone profile (V8PRO) algorithm, which makes use of a number of shorter wavelengths for estimating ozone amounts in the upper layers of atmosphere, the V8TOZ for NOAA OMPS S-NPP and N20 makes use of 12 discrete channel wavelengths [308.7nm, 310.8nm, 311.9nm, 312.6nm, 313.2,nm 314.4nm, 317.6nm, 322.4nm,
331.3nm, 345.4nm, 360.2nm, 372.8nm] in the retrieval algorithm. Of those 12 wavelengths, the two wavelengths (317.6 and 331.3 nm) are directly used to derive total ozone, while other channel wavelengths are used to make error corrections from aerosols, profile shapes, clouds, sun-glint as well as to provide atmospheric $SO_2$ estimates for volcanic eruptions (Niu et al., 2020; Yang et al., 2009).

### 2.1 Forward Model

Because the OMPS sensors only measure BUV from the top of atmosphere (TOA), radiative transfer forward model results must be included in the algorithm under various ozone amounts and vertical distribution conditions as well as geometrical properties. To minimize computer time, the TOA radiances are computed by interpolation and adjustment from a pre-computed radiance table, which is created using the TOMRAD radiative transfer code (Caudill et al., 1997). This table consists of five variables: $I_0$, $I_1$, $I_2$, $I_R$ and $S_b$. Using these five variables one can calculate the TOA radiance I with the following formula:

$$I = I_0(\theta_0, \theta) + I_1(\theta_0, \theta)\cos\phi + I_2(\theta_0, \theta)\cos 2\phi + \frac{RI_R(\theta_0, \theta)}{(1 - RS_b)} \qquad (1)$$

where, the first three terms together constitute the purely atmospheric component of the radiance, unaffected by the surface. This component, which we will refer to as $I_a$, is a function of solar zenith angle $\theta_0$, satellite zenith angle $\theta$, and $\phi$, the relative azimuth angle between the plane containing the sun and local nadir at the viewing location and the plane containing the satellite and local nadir. The last term in Eq.1 provides the surface contribution, where, $RI_R$ is the once-reflected radiance from a
Lambertian surface of reflectivity R, and the factor $(1 - RS_b)^{-1}$ accounts for multiple reflections between the surface and the overlying atmosphere.

Since the OMPS-NM uses a CCD array, there are essentially thousands of independent detectors. This means that the products from each cross-track FOV are derived from their own set of detectors. Different from NASA OMPS S-NPP Level 1 product, in which the CCD readout is kept split at the center with 36 cross-track measurements per swath, both OMPS S-NPP and N20
V8TOZ products at NOAA have a 110° cross-track FOV with 35 cross-track bins. While OMPS S-NPP has a 0.27° along-





track slit width, corresponding to a 50x50 km resolution at the Earth's surface, OMPS N20 has a 0.09° along-track slit width with a 50x17 km resolution. Both of them provide instantaneous coverage of a 2800-km-wide swath at the Earth's surface.

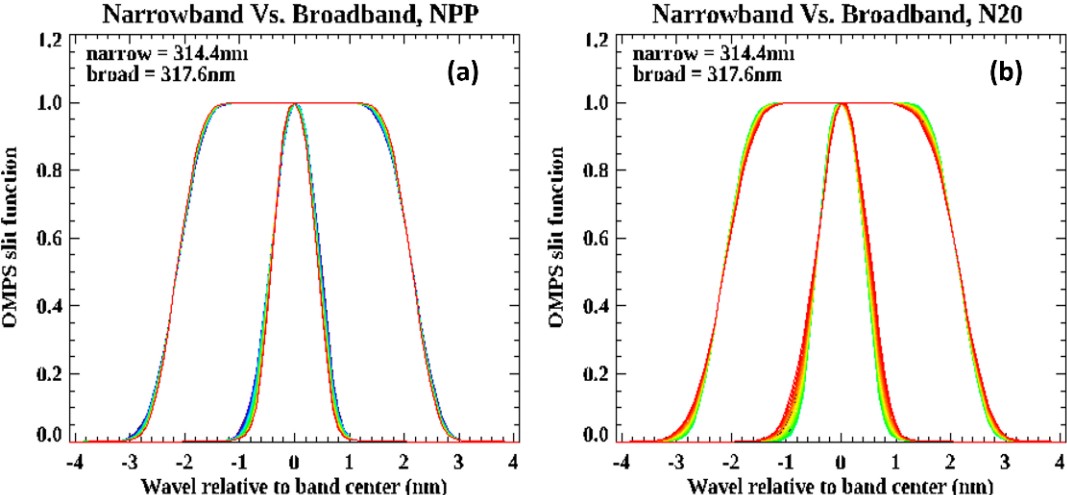

**Figure 2.** (a) OMPS NPP pre-flight slit function at 314.4 nm narrowband center and 317.6nm broadband center for 35 cross-track positions,
with colors representing different cross-track positions from 1 (blue) to 35 (red). (b) Same as (a) but for OMPS N20.

Due to the differences in their CCD detectors and bandpasses, we generated N-value [N-values are defined as -100*log$_{10}$(Radiance/Irradiance)] look up tables from the TOMRAD radiative transfer code for both OMPS S-NPP and N20. Those tables are computed for 10 solar zenith angles, 6 satellite zenith angles, 4 surface pressures, 12 channel wavelengths, 21 *a priori* ozone profiles, and 35 cross-track positions. The main differences in the tables between S-NPP and N20 are their

instrument band-pass functions which differ for different instruments. In this study, we use broad bandpasses for the six longer wavelength channels and narrow bandpasses for the shorter wavelength channels. Fig. 2 shows preflight slit functions at 314.4 nm narrowband center and 317.6 nm broadband center for 35 cross-track positions for OMPS S-NPP and N20. The slit functions for the other five narrow bandpasses used for channel centers shorter than 314.4 nm channel wavelength, and five broad bandpass used for channel centers longer than 317.6 nm wavelength are not shown here due to the similarity. The slit

functions provide key information for the spectra convolved values of the ozone absorption cross-sections. The OMPS detectors make measurements every 0.42 nm from 300 to 380nm with around 1.0-nm full width at half maximum (FWHM) resolution. The OMPS V8TOZ products in NOAA have recently switched to use broader bandpasses for the six longer channel wavelengths in the algorithm. As shown in Fig. 2, the broadband slit function for 317.6 nm channel appears to become very flat around its peak values for both S-NPP and N20. Those slit functions are the aggregated narrowband slit functions from

eleven adjacent wavelengths surrounding 317.6 nm in the measurements. Calculations indicate that the differences of FWHMs for narrowband slit function between S-NPP and N20 are about 0.75%, while those differences for broadband slit function with the FWHMs close to 4.3 nm are negligible. That suggests using broader bandpass should provide a tendency to reduce



retrieval biases from potential biases in ozone absorption cross-sections. We will address further advantages of using broader bandpass wavelengths in retrievals later.

## 2.2 Inverse Model

The basic OMPS V8 algorithm uses just two wavelengths to derive total column ozone: a strong ozone absorption wavelength (317.6nm) to estimate total ozone, and a weakly absorbing wavelength (331.3 nm) to estimate an effective surface and cloud reflectivity. The formula of the forward model (Eq.1) indicates that BUV radiance can be estimated if we know the underlying reflectivity with a given total ozone associated with the ozone profile. Since we only have measured radiance/irradiance ratios, to solve this TOMRAD radiative transfer equation for a proper estimation of total ozone, an iteration approach has to be employed, i.e., by starting with a nominal total ozone estimate and then recalculating the reflectivity using total ozone provided by the 317.6 nm shorter wavelength. The process is repeated if the estimated reflectivity changes significantly. The inverse process assumes that the effective reflectivity estimated from 331.3 nm wavelength has little dependence on wavelength. This assumption is pretty robust since ozone absorption at 317.6 nm wavelength is much larger than that for 331.3 nm wavelength and the channel separation is less than 14 nm.

## 2.3 Aerosol Index

In the inverse model, the effective reflectivity is estimated from 331.3 nm wavelength, and taken to be insensitive to change of wavelengths. However, when radiances encounter aerosol particles, the interactions will influence the reflectivity, and result in apparent dependence of reflectivity on wavelengths. That is why the AI was introduced in the V8TOZ algorithm for correcting total ozone retrievals, and it turned out to be a very useful product for monitoring environmental change. The aerosol index for OMPS S-NPP and N20 is defined as the difference between the measured (includes aerosols effects) spectral contrast of the 360.2 and 331.3 nm wavelength radiances and the contrast calculated from the radiative transfer theory for a pure molecular (Rayleigh scattering) atmosphere. Since the calculation of the radiance for 360.2 nm wavelength uses reflectivity derived from the 331 nm measurements, the Aerosol Index can be simply defined as:

$$AI = 100 \, log_{10} \left( \frac{I_{360\_meas}}{I_{360\_calc}} \right) \tag{2}$$

Torres and Bhartia [1999] showed that a simple linear relationship exists between the Aerosol Index values and the total column ozone error, and that the slope of this relationship varies with slant path ($sec\theta_0 + sec\theta$). A correction is applied using the AI values and tabulated value of these slopes in the OMPS V8TOZ algorithm. The positive AI values are associated with UV-absorbing aerosols, mainly mineral dust, smoke and volcanic aerosols, and the negative values are associated with non-absorbing aerosols (for example, sulfate and sea salt particles) from both natural and anthropogenic sources (Torres et al, 1998) with sizes less than 0.2 microns.



## 2.4 Broader Bandpass Approach vs. Narrowband Approach

When we average radiance and irradiance from eleven adjacent spectral channels in our retrieval, we expect the measurement noise as well as biases and uncertainties from wavelength shifts, interpolation from measurements to algorithm channel
wavelengths, Ring effects (inelastic scattering including Telluric contributions not present in the radiative transfer forward model), and stray light correction uncertainties will be greatly reduced. The improvements for errors in the wavelength scale are best for channels with relatively flat radiances. The 318 nm channel is in a region with a linear gradient in radiances versus wavelength, so the benefit of reduced sensitivity to wavelength shifts is not present. The interpolation errors are related to the effective broadening of the bandpasses when measurements are used to estimate the signals at intermediate points. By using
broader bandpasses to start, the interpolation distances in wavelength space are a factor of eleven smaller relative to the bandpass width than with single measurements. One potential caveat of applying broader bandpasses in the algorithm is that it would slightly weaken the spectral contrast for retrievals. Statistical analysis of the total ozone retrievals shows that this weakness is negligible. We do not use the broader bandpasses for the six shorter channels as they are used to determine estimates of atmospheric $SO_2$ by using a follow-on algorithm which makes use of the smaller-scale spectral features of $SO_2$
absorption.

We made one month (March 2020) runs of V8TOZ for both OMPS S-NPP and N20 with zero soft-calibration based on both narrowband radiances and broadband radiances. The 31 days' pixel level ozone deviations were averaged as shown in Fig. 3. Those deviations appear to contain two parts. One is from the real ozone gradations in space, and the other is from retrieval biases with various sources. Because the true ozone patterns are independent of sensors and algorithms, comparing the
deviations would be able to show the magnitude of error biases from retrievals. It is expected that if one algorithm can generate a more smooth and homogeneous retrieval with less noises, the deviation will be smaller. Thus, this deviation can be referred to as homogeneity deviations, an averaged deviation of pixel ozone with those pixels of its neighboring cross-tracks and scans.







**Figure 3.** The March 2020 along-orbit homogeneity deviations of retrieved total column ozone in DU for 35 cross-track positions for
narrowband (left panel) and broadband (middle panel) of OMPS S-NPP (top) and OMPS N20 (bottom), as well as the differences of
homogeneity deviation between narrowband and broadband (right panel). Those are along orbit ozone retrieval variability with respect to
different cross-tracks for both OMPS S-NPP and N20 based on narrowband wavelength approach as well as based on broadband. It is simply
an averaged deviation of pixel ozone with its neighboring cross-tracks as well with its neighboring scans at the same cross-track position.
Those absolute deviation values were binned at 0.5° intervals according to cross-track #1 solar zenith angle. So, approximately 400 scans of
NPP NM and 1200 scans of N20 NM were averaged onto 360 SZA intervals for one orbit.

The patterns of homogeneity deviation appear to be pretty similar between OMPS S-NPP and N20, which show the Northern

hemisphere has larger variation than Southern hemisphere, and the equatorial and lower latitudinal regions exhibit lowest

ozone variability. Those deviation structures correspond well with the natural global ozone patterns except for some

uncommon features. The deviation for OMPS S-NPP appears to be larger than N20, which may be mainly due to the fact that

the FOVs of S-NPP are three times wider than those for N20. The homogeneity deviation also shows apparent association with

ozone slant column density (SCD). That suggests the V8TOZ algorithm tends to have systematic biases as SCDs get larger.





There are striping structures in both S-NPP and N20. Those features mainly come from measurement calibration errors associated with cross-track positions. By subtracting the homogeneity deviation of narrowband retrievals from broadband retrievals (see Fig. 3, left panel), we see apparent bias reduction when using broadband wavelengths, especially for N20. The

bias reduction with broadband wavelengths did not show noticeable improvement for retrievals at high SCD regions. That suggests that broadband approaches could make apparent minimization of measurement noises, but would have limited influences on error biases from the algorithm itself.

## 3 Soft-calibration for both OMPS S-NPP and N20

The purpose of soft-calibration is to make retrieved variables such as total column ozone and aerosol index be close to their

true states. As mentioned before the OMPS satellites use a CCD array, and the radiances were binned onto cross-track FOV with 35 cross-track positions. For the V8TOZ applications, it is as though both OMPS S-NPP and N20 contain 12x35 individual instruments, which have their own measurement and calibration errors from various sources. In order to make consistent retrievals with close to "truth" values, we have to conduct a radiance adjustment for each of the 105 (318 nm, 331 nm and 360 nm channels by 35 cross-track) individual instruments from S-NPP and N20, and to make them have the same unbiased

performance. Similar approach to obtain soft-calibration adjustments was addressed in other works (Bak et al., 2017).

There are two key assumptions for our soft-calibration: One is that the natural patterns of ozone and aerosol index are homogeneous with little dependence on cross-track positions. So that if we do an averaging of ozone and aerosol index at different cross-track positions based on a certain amount of data in a region, those averaged ozone and aerosol index should be close to the same values as expected. The second assumption is that those cross-track related measurement errors are

consistent. That is the magnitude of errors caused by biases in radiance or in wavelength registration should have a similar pattern along an orbit.

### 3.1 Data

Two months' data were used in estimating our soft-calibration for NOAA OMPS S-NPP and N20. The March 2020 data are mainly used to generate soft-calibration adjustment tables, while the September 2020 retrievals are used only for comparison

and validation to verify the fidelity of the soft-calibration, due to that the September data are fully independent of building the soft-calibration tables. Data from the NASA OMPS Nadir Mapper Suomi NPP NMTO3-L2 from NASA OZONE & AIR QUALITY website are used for calibration and comparison. We selected NASA's retrievals instead of ground-based observations for calibration simply because the NASA OMPS S-NPP V8TOZ retrievals have been well validated and generally used for comparison with ozone retrievals from other satellites. The NASA products were calibrated to agree with NOAA-19

SBUV/2 total ozone and with cross track adjustments to the effective reflectivity from ice radiance studies (McPeters et al., 2019). We use these products to tie the OMPS retrievals to the earlier ozone record. Since the NOAA operational OMPS



V8TOZ products share the same algorithm with the same measurements as NASA, it is easier to make these two products agree with each other. However, because NASA binned the CCD array in 36 FOVs, which is different from our 35 FOVs, and NASA and NOAA are slightly different in processing SDR data, employing soft-calibration directly from NASA OMPS V8TOZ as NOAA adjustment tables even for the same narrowband approach will not work properly.

## 3.2 Method

The V8TOZ output contains a variety of useful parameters in addition to the total column ozone, effective reflectivity, and aerosol index estimates. In particular, the retrieval sensitivities, dy/dx, of the forward model predicted measurement, y, to a retrieved parameter, x, can be used to give soft calibration estimates of the N-value changes to remove reflectivity and ozone bias. If you want to increase the effective reflectivity, R, and the total column ozone, $\Omega$, by $\Delta R$ and $\Delta \Omega$ then you should increase the N-values by

$$\Delta N_{318} = \Delta R \frac{dN_{318}}{dR} + \Delta \Omega \frac{dN_{318}}{d\Omega} \tag{3}$$

$$\Delta N_{331} = \Delta R \frac{dN_{331}}{dR} + \Delta \Omega \frac{dN_{331}}{d\Omega} \tag{4}$$

where $dN_w/dR$ is the rate of change of the N-value, defined as $-100*\log_{10}(Radiance/Irradiance)$, $N_w$, for wavelength, w, with respect to changes in the effective reflectivity, R, and $dN_w/d\Omega$ is the rate of change of the N-value, $N_w$, for wavelength, w, with respect to changes in the total column ozone of $\Omega$ in Dobson Unit (DU). Those are two key equations used to estimate soft-calibration parameters for the 318 nm ozone channel and the 331 nm reflectivity channel. We calculate the soft-calibration values for all the other channels using pretty much the same equation by assuming that there is no change of effective reflectivity with respect to different wavelengths, and the calculated N-value residuals, which represent the differences between measurements and the forward model using the retrieval values, should keep the same value for 35 cross-track positions.

In Eq.3 and Eq.4, $\Delta R$ represents the departures of true reflectance with the retrieved effective reflectivity without soft-calibration adjustment for 35 cross-track positions, and $\Delta \Omega$ represents the differences of real (true) total column ozone with respect to retrievals without soft-calibration. We assume that the monthly averaged effective reflectivity and total column ozone from NASA retrievals are close to real states. Then, to calculate soft-calibration values for $\Delta N_{318}$ and $\Delta N_{331}$, we only need to know the values of $dN_{318}/dR$, $dN_{318}/d\Omega$, $dN_{331}/dR$, and $dN_{331}/d\Omega$. Those four parameters are outputs from V8TOZ algorithm varying with respect to estimated ozone and reflectivity. So, an iterative process is needed to achieve the right values for these two soft-calibration parameters.

In order to obtain a universal soft-calibration table working everywhere, we choose two regions to conduct the adjustment parameters. We derive $\Delta R$ over the Equatorial Pacific box defined by 20°S to 20°N, 100°W to 180°W because of its benign and seasonally stable conditions, such as dark ocean surface and low UV-absorbing aerosol loading. The one-percentile effective reflectivity values are used to represent a relatively low reflectivity with limited cloud influence. That means, for a set of reflectivity data values, 99% of the values are larger than this one-percentile reflectivity; correspondingly, the standard





median is the 50-percentile value. We derive ΔΩ over lower latitude land regions between 25°S to 25°N. The averaged ozone values were computed from cloud-free pixels of best ozone retrievals. The best ozone has been adjusted with aerosol loading

which was also flattened over the 35 cross-track positions for a month averaging. This is not always true of the V8TOz algorithm. It is difficult to obtain a real association of AI patterns with TCO with respect to different cross trucks and satellite viewing angles. We here make a simple assumption that the natural AI would not be a function of cross-track and SVA, except in the presence of sun glint. One obvious advantage of choosing cloud-free land retrievals for deriving soft-calibration is that it reduces potential complications from clouds and sun glint.

## 3.3 Calibration results

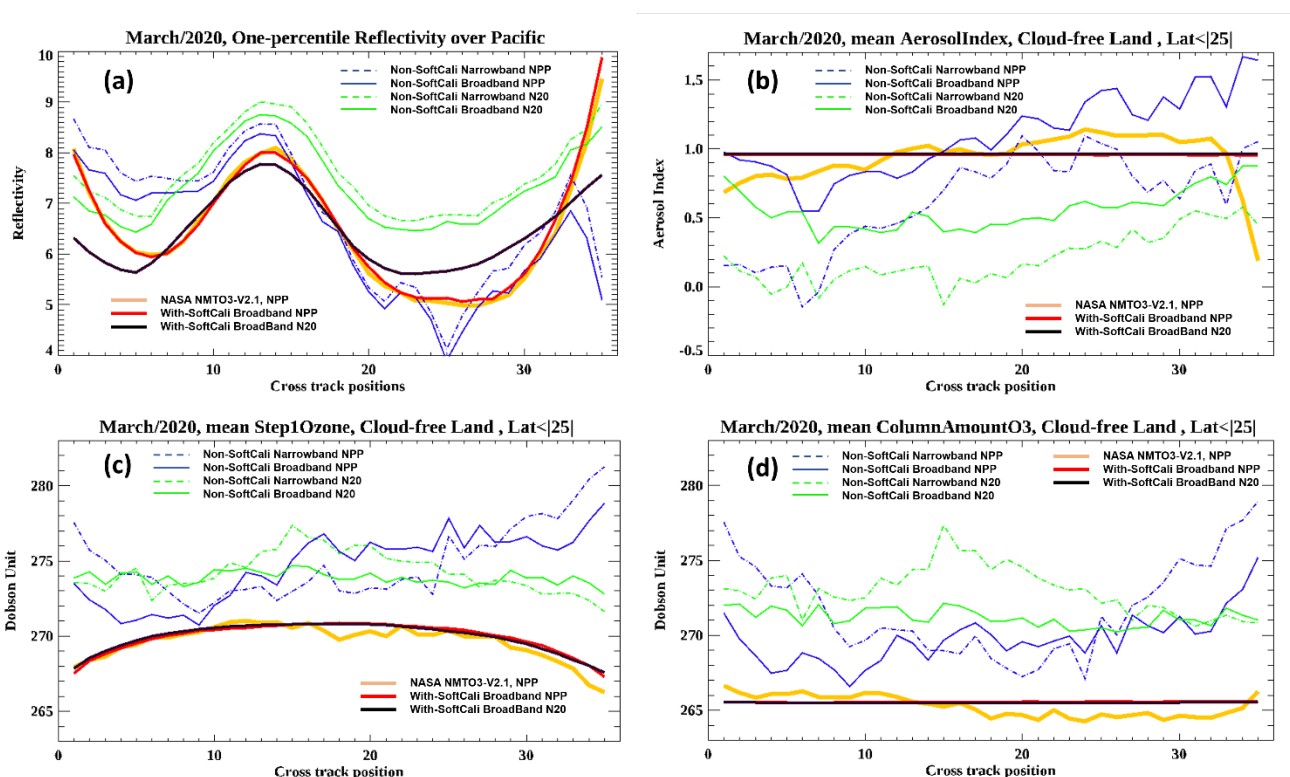

**Figure 4.** The March 2020 one-percentile reflectivity over the Pacific box (a), mean aerosol index (b), mean Step-1 ozone (c) and mean best ozone retrieval (d) for 35 cross-track positions. The yellow bold lines represent NASA NMTO3-V2.1 OMPS S-NPP retrievals, which were used as references for soft-calibration; the blue lines represent OMPS S-NPP retrievals of narrowband approach (dash) and broadband

approach (solid) before soft-calibration; the green lines represent OMPS N20 retrievals of narrowband approach (dash) and broadband approach (solid) before soft-calibration; the bold lines represent OMPS S-NPP retrievals of broadband approach (red) and N20 retrievals of broadband (black) after soft-calibration.

Figure 4-a shows one-percentile reflectivity over the Pacific box for March 2020. We put the one-percentile reflectivity from NASA NMTO3-v2.1, OMPS S-NPP on the same figure for comparison. The NASA reflectivity, which has been adjusted

based on the darkest pixels over land, shows a wave-like pattern along cross-tracks with the lowest values about 5% around



cross-track position #25. The wave-like reflectivity pattern for OMPS S-NPP is typical over oceans with a sun-glint hump from cross-track positions #7 to #19, and viewing angle effects from haze, aerosol and fair-weather cumulus clouds that are frequently present at very low altitudes. At higher viewing angles, sky glint can also become a larger contribution. The relatively larger FOVs at higher view angles will make fully clear scenes less likely. The one-percentile reflectivity before

soft-calibration appear to have cross-track related structures come from biases in the level-1 SDR data for both OMPS S-NPP and N20. The unadjusted OMPS S-NPP results show larger variations of reflectivity than the variations for N20, especially in the eastern half of cross-tracks. The reflectivity from broadband approach is generally slight lower with less variation than narrowband approach.

     We forced the broadband reflectivity (red line in Fig. 4-a) of NOAA OMPS S-NPP to be in agreement with NASA NMTO3-

v2.1 (yellow line). As by design, we not only made the reflectivity nearly the same for different platforms but also removed apparent biases associated with cross-track positions. As shown in Fig. 4-a, the reflectivity of N20 for both narrowband and broadband appears to be close to the expected patterns with sun-glint hump and rises at high viewing angles. It is quite reasonable to consider that this wave-like reflectivity pattern of N20 is close to the real one. In order to make the NOAA OMPS N20 retrievals in agree with the retrievals from OMPS S-NPP, we forced the averaged one-percentile reflectivity of N20 to be

the same as the mean one-percentile of S-NPP, and removed all those cross-track biases through smoothing (see black line at Fig. 4-a). That is, we trust the relative calibration for OMPS N20 from the ground-based characterizations. Notice that this gives us two different patterns for the cross-track (viewing angle) one-percentile effective reflectivity dependence over the equatorial Pacific.

     Figure 4-b shows monthly mean aerosol indexes over low latitude land between 25°S and 25°N for all those cloud-free pixels.

To make the NOAA OMPS S-NPP/N20 retrieved aerosol index in agreement with NASA retrievals, we averaged 12 cross-tracks' NASA OMPS S-NPP NMTO3-2.1 AI values (yellow line) close to nadir as the true aerosol state. It is not necessarily true that, in nature, the mean state of the aerosol index as defined with the simple V8TOz cloud model should be flattened with no dependence on cross-track positions. Any cross-track related ups and downs of the retrieved aerosol indexes represent biases from various sources in addition to instrument calibration. Without soft-calibration adjustments, the retrieved aerosol

indexes show apparent variation with respect to different cross-track positions for both OMPS S-NPP and N20. The retrieved aerosol indexes based on broadband approach appear to have less variability than those from narrowband approach, indicating some advantage of using broader bandpass channels. There is about 0.5 difference in retrieved aerosol indexes between OMPS S-NPP and N20 before soft-calibration, which may imply some relatively large biases in level-1 SDR data for longer wavelengths. After soft-calibration adjustment, we see that the averaged aerosol index retrievals are virtually the same as the

expected true aerosol states, with no differences between NOAA OMPS S-NPP (red line) and N20 (black line) and no associations with cross-track positions. The reflectivity model will have some view and solar zenith angle dependent errors and the aerosol index for a given atmospheric aerosol will vary due to the wavelength dependence of these errors as well as the view and solar zenith angle channel sensitivities to the aerosol layer height (Haffner et al. 2019).



Figure 4-c has the same conventions as Fig. 4-b except that all the lines represent monthly averaged Step-1 total column
ozone instead of aerosol index. As expected, without soft-calibration adjustments, the retrieved Step-1 ozone values show
apparent variation with respect to different cross-track positions for both OMPS S-NPP and N20, and the retrievals based on
broadband approach appear to have less variability than those from narrowband approach. The retrieved Step-1 ozone with
soft-calibration adjustment shows a pretty smoothing curve-like pattern for both S-NPP (red line) and N20 (black line). The
magnitude and curve-like features are in agreement with those from NASA Step-1 ozone retrievals (yellow line). The curve-
like shape seems to be true with the current version 8 total column ozone algorithm since both NOAA and NASA retrievals
have this feature. The reason for those potential biases at the higher satellite viewing angles may be that: the final ozone
retrievals were adjusted to remove aerosol effects, and those adjustments are virtually scale factors associated with aerosol
indexes. It is likely those scale factors were not able to precisely account for the influences of slant column density (SCD), and
cause up to 1% bias at extreme satellite viewing angles for Step-1 ozone. The retrieval algorithm is also sensitive to differences
between the true profiles and the set of standard profiles used to construct the instrument tables. Further, these errors will vary
with the retrieval sensitivity (layer retrieval efficiency factors) to the profile shape and that in turn varies with the viewing and
solar zenith angles.

With the same conventions as Fig. 4-b and 4-c, Fig. 4-d shows monthly mean final (Step-3) total column ozone retrievals
over low latitude land between 25°S and 25°N from all cloud-free pixels. A similar idea as was used in making soft-calibration
for aerosol index, we treat the averaged 12 close-to-nadir cross-tracks' NASA OMPS S-NPP NMTO3-2.1 final ozone values
(yellow line) as the true ozone state, and adjusted the radiances to make NOAA OMPS S-NPP and N20 total ozone retrievals
with broadband approach the same as this "true" ozone state for each of those 35 cross-tracks. Since the final ozone retrieval
involves initial estimates of ozone and effective reflectivity with 318 and 331nm channel radiances, as well as radiances from
other wavelengths for aerosol and ozone profile shape adjustments, an iterative approach was employed in our process. As
shown in Fig. 4-d, before soft-calibration, both OMPS S-NPP and N20 exhibit apparent higher than "true" state ozone values,
and the retrievals from narrowband approach appear to have larger retrieved ozone with more variability along 35 cross-tracks.
Like the retrieved aerosol index with adjustment, the monthly averaged final total column ozone after soft-calibration is
flattened with respect to different cross-tracks, suggesting intra cross-track biases are virtually gone by adjusted radiances. The
final ozone retrievals for both NOAA OMPS S-NPP (red line) and N20 (black line) are very close to the "true" ozone state
derived from NASA OMPS S-NPP, indicating that the soft-calibration parameters generated for NOAA OMPS are quite
robust. We did not plot out the generated 12 (wavelengths) x 35 (cross-tracks) adjustment values here for OMPS S-NPP and
N20 since they are saved in the NOAA OMPS V8TOZ Environmental Data Record (EDR) NetCDF products.

## 3.4 Verification results

Figure 4 shows the one-percentile reflectivity over Pacific, the monthly averaged aerosol index, Step-1 ozone and final (Step-
3) ozone retrievals over cloud-free land before and after soft-calibration adjustment for NOAA OMPS S-NPP and N20.



However, since the broadband retrievals for the entire month of March are involved in the generation of soft-calibration parameters, it is possible that biases from the data will be forced to add to the soft-calibration. For example, some retrievals based on biased measurement may not be screened out from statistics, or the true states of ozone and aerosol index were not flattened with the assumed values due to insufficient data pool. We really need to see retrievals in other seasons that are fully

independent of calibration to verify the stability and robustness of our soft-calibration.

**Figure 5.** The September 2020 one-percentile reflectivity over Pacific (a – top left), mean aerosol index (b – top right), mean Step-1 ozone (c – middle left), mean best (Step-3) ozone retrieval (d – middle right) and one-percentile reflectivity over land (e – bottom left) for 35 cross-

track positions. The yellow bold lines represent NASA NMTO3-V2.1 OMPS S-NPP retrievals; the thin lines represent OMPS S-NPP



retrievals of broadband approach (blue) and N20 retrievals of broadband (green) before soft-calibration; the bold lines represent OMPS S-NPP retrievals of broadband approach (red) and N20 retrievals of broadband (black) after soft-calibration.

Figure 5(a-d) shows the same statistics as Fig. 4 except instead of March 2020, the one-percentile reflectivity, aerosol index and ozone are all from retrievals in September 2020, a month that is independent of the soft calibration adjustment month. We
did not put any of the retrievals with narrowband here since the narrowband approach is no longer used in NOAA OMPS operational V8TOZ retrievals. We added one extra plot (Fig. 5-e) here for better understanding of differences in behavior of effective reflectivity over ocean and over land. As expected, the variation features of one-percentile (Fig. 5-a), aerosol index (Fig. 4-b), Step-1 ozone (Fig. 5-c) and final ozone (Fig. 5-d) without soft-calibration adjustment are very similar to those in Fig. 4 for both OMPS S-NPP and N20. Although the mean states were shifted due to seasonal change, the high similarity for
those variability structures, and almost the same departures in retrievals between OMPS S-NPP and N20 with respect to cross-tracks suggest that: the OMPS instruments are very stable, and the remaining intra cross-track biases are mainly from systematic errors in measurements, which are the main sources for persistent biases in retrievals.

The September one-percentile reflectivity with adjustments (Fig. 5-a) appears to be very close to those in March with expected sun-glint hump and rises at high viewing angles. For OMPS S-NPP, reflectivity from NASA (yellow line) and NOAA (red
line) are almost the same suggesting both products have kept their relative calibration. The N20 reflectivity (black line) with adjustments seems not as smooth as that from the calibration in March, but no noticeable cross-track related biases exist, indicating the soft-calibration for NOAA OMPS N20 reflectivity is also robust. Similar to aerosol index in March, the adjusted monthly mean aerosol index for September (Fig. 5-b) is close to 1 for both OMPS S-NPP (red line) and N20 (black line), very slight curve-like shape may indicate somewhat non-linearity, and very small variation in terms of different cross-tracks may
come from noises in statistics. Like Step-1 ozone in March, the adjusted monthly mean ozone for September (Fig. 5-c) shows similar curve-like patterns for both NOAA OMPS S-NPP (red line) and N20 (black line), with slightly more bending at high viewing angles. Since the shape and magnitude of NOAA OMPS Step-1 ozone are quite similar to that of NASA OMPS S-NPP, it is most likely those errors are from non-linearity associated with SCD as we mentioned before. Since we forced the monthly mean of final ozone retrievals to be flattened in March by soft-calibration, it is expected that we should have a flatter
ozone for the September retrievals. However, the final ozone retrievals for both NOAA OMPS S-NPP (red line) and N20 (black line) as well as NASA OMPS S-NPP (yellow line) show similar curve-like features with potential errors reaching up to 1% at very edged pixels in a swath. Those errors most likely come from non-linearity associated with slant column density and profile shape interactions that are not accounted precisely in the algorithm. Nevertheless, the soft-calibration parameters for most pixels are quite robust which are capable of producing high quality V8TOZ retrievals with high agreement with
NASA OMPS retrievals. Systematic differences between the true ozone profiles and the standard profiles used in the radiative transfer tables can also produce cross-track dependencies as the layer retrieval efficiency varies with both view angle and solar zenith angle.

Except for the extreme off-nadir cross-tracks positions, both OMPS S-NPP and N20 show relatively flat reflectivity over land (Fig. 5-e), the sun-glint effects which causes a significant hump at the ocean are gone. Reflectivity of S-NPP shows a very



slight downward trend from west to east, while N20 shows a somewhat upward trend. In the east part of cross-tracks, the difference of effective reflectivity between S-NPP and N20 is about 0.5, which is very close to the magnitude of difference at those cross-track positions when we built the soft-calibrations using one percentile reflectivity over the Pacific. Maybe the true reflectivity over those cross-tracks is in the middle of the retrieved reflectivity between S-NPP and N20. For the extreme off-nadir cross-tracks positions, the reflectivity of OMPS S-NPP still shows bias attribute to sky-glint effects at the higher viewing

angles which suggest that the post soft calibration pattern over the ocean was too large. However, the reflectivity of OMPS N20 appears to have downward trend in the east. It is opposite to the OMPS S-NPP with the departure magnitude close to that when we made the soft-calibrations. More studies are needed to decide if this is true reflectivity for OMPS N20 at those off-nadir cross-track positions. One potential reason for that lower reflectivity may be that: because of the much smaller FOVs for N20 compared to S-NPP, one expects it to be harder (less likely) to have clear scenes as the area of the FOV increases. It is

also the case that the simple reflectivity model used in the V8TOz may have SZA and SVA dependent retrieval errors.

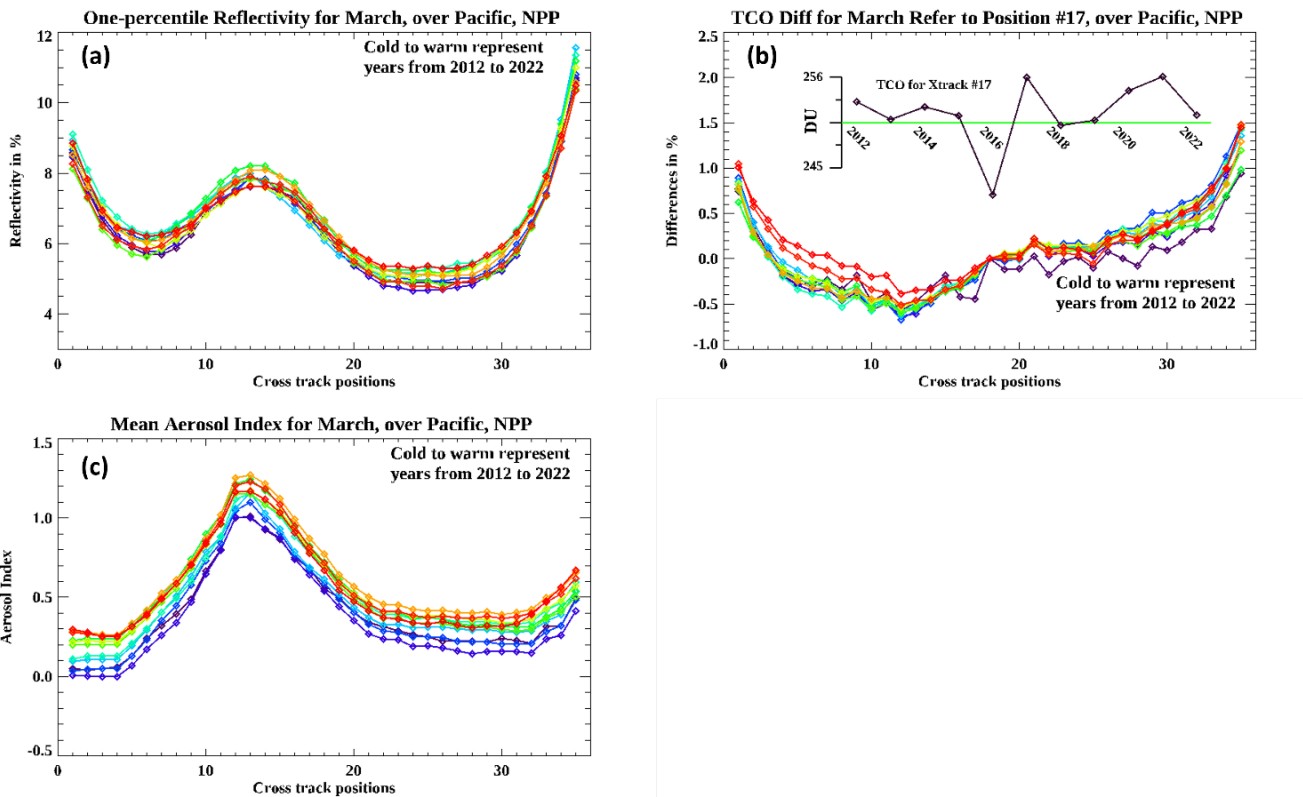

**Figure 6.** Cross track patterns for the Equatorial Pacific for March for all 11 years (2012 to 2022 – Cold to Warm) of the S-NPP record. Fig. 6-a shows the cross-track dependence of the one-percentile Reflectivity. Fig. 6-b shows the averages versus cross-track for the total column

ozone. The inset time series shows the averages for cross-track position #17. Fig. 6-c shows averages versus cross-track for the Aerosol Index.



To further check the performance of the soft calibration, we examined the cross-track and absolute dependence of the effective reflectivity, total column ozone and aerosol index values over the equatorial Pacific box for March for all 11 years in the S-NPP data record. Fig. 6-a shows that the cross-track pattern for one-percentile reflectivity is very stable year-after-year, and the absolute values are stable at the 1% level. While we have used Effective Reflectivity values from comparison over the same time periods in 2020 to make our soft calibration adjustment estimates to force agreement between S-NPP and NOAA-20, this suggests that there is some value to comparing the absolute results over time as a stability check both for cross-track patterns and absolute values. Fig. 6-b shows that the cross-track pattern for the total column ozone over the Equatorial Pacific box is also stable year-after-year. The values are given relative to cross-track position 17. The average TCO values for position 17 over the 11 years vary by 7% with no specific trend as shown in the inset time series. One expects variations in TCO for this region related to dynamics such as the Quasi-Biennial Oscillation. Fig. 6-c shows that the cross-track pattern for the aerosol index over the Equatorial Pacific box is also stable. There is a trend totaling approximately 0.3 over the 11 years. This is consistent with the S-NPP differences in the instrument throughput changes for the 331 nm and 360 nm channels as shown in Figure 1 and the formulation of the aerosol index.

## 4 Errors and Uncertainties versus Latitude

As mentioned in Section 3, soft-calibration may suffer from sampling biases as well as non-linearity and slant column density issues. In addition, there are issues of measurements in geophysical properties of Earth surface, geometric inaccuracy and radiance and wavelengths inconsistency along orbit with possible seasonal variations. All those issues are likely to bring errors and uncertainties in our final retrievals. Because of the complexity of those biases and the associations with various steps in the retrieval algorithm, explicitly addressing the relationships and magnitudes of the effects in the algorithm is difficult. Since the final retrievals of ozone and aerosol index are the main products of OMPS S-NPP and N20, we will focus on analyzing the spatial and temporal variations of the differences for those two variables. If those detected errors are persistent with respect to cross-track positions, solar zenith angles and seasons, then, we would think those errors are consistent and systematic, and capable of removing by correction with latitudinal and cross-track scale factors or adjustments.

### 4.1 Error detection

Both OMPS S-NPP and N20 make ~14 sunlit orbital measurements in one day with ~50 minutes' difference at equator for consecutive orbits and generate full global coverage retrievals. The two platforms have the same local Equator crossing times but are situated 180° apart in their orbits. That means, for equatorial areas we will have at least two measurements from OMPS S-NPP and N20 with 50 minutes' difference in measuring time. For middle and high latitude regions, due to the geophysics of the Earth's shape, there will be more than two overlapped measurements. Because S-NPP and N20 are from the same series



of NOAA OMPS satellites, and the same algorithm was used for retrievals, the gridded retrievals combined those two instruments should be able to provide further internal (or cross satellite) comparisons for estimating error biases.

### 4.1.1 Biases in retrieved AI

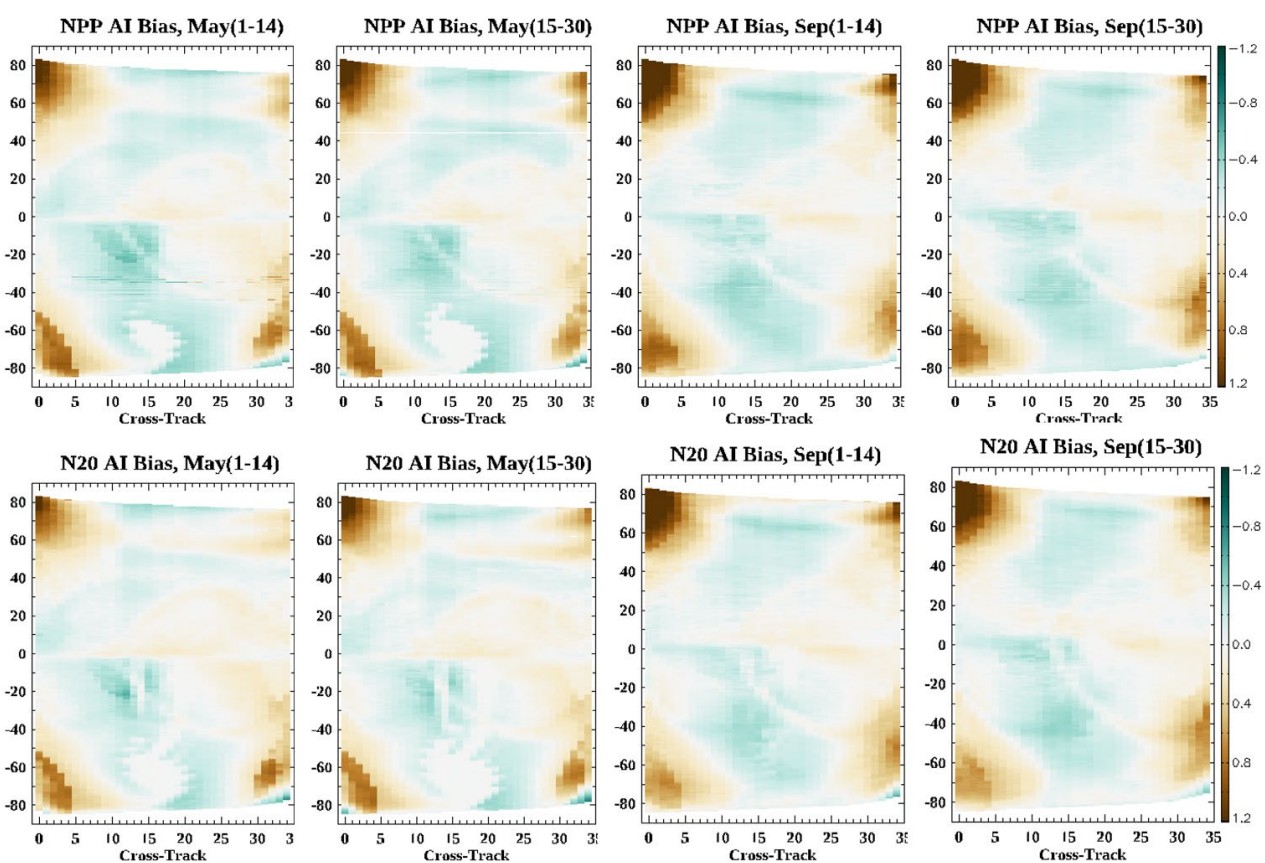

**Figure 7.** The along orbit retrieved aerosol index biases associated with 35 cross-tracks for NOAA OMPS S-NPP (upper panel) and OMPS N20 (bottom panel). The left two plots represent the averaged aerosol index biases for the first and second half of May 2020, while the right two plots represent the averaged half-month aerosol index biases for September 2020.

Figure 7 shows retrieved aerosol index biases along orbit with respect to 35 cross-track positions after soft-calibration for both NOAA OMPS S-NPP and N20. The biases were calculated as follows: First, the daily 0.5 latitude x 0.5 longitude degree gridded aerosol index dataset was established based on combined aerosol index retrievals from OMPS S-NPP and N20. In building this gridded dataset, distance weight was applied to those pixels selected to generate the gridded values. Only high-quality pixel retrievals with no contamination from sun-glint and no influence from larger $\sec(\theta_0)$ or $\sec(\theta)$ retrievals, and within one degree (latitude/longitude) searching radius, were chosen for averaging. Second, the bias of aerosol index for each individual pixel was estimated by subtracting the retrieved aerosol index from gridded value in the same grid, and thus we will



have a pixel level bias dataset of aerosol index with the same dimensions of retrievals as OMPS S-NPP and N20. We averaged those biases in terms of solar zenith angles/scans and cross-track positions for 15 days, and generated the along orbit bias patterns as shown in Fig. 7.

Two months' data (May and September 2020) were employed to illustrate the features of retrieved aerosol index biases along orbit. The bias patterns appear to be very similar and persistent in terms of different satellites and different seasons. Because

geolocations of pixels at the same solar zenith angle differ a lot between May and September, the high similarity indicates the associations of biases with geolocation are limited compared to solar zenith angle. Since intra cross-track biases and scale differences for both OMPS S-NPP and N20 have been mostly removed by soft-calibrations over equatorial regions, the biases detected in Fig. 7 are likely associated with either inconsistent biases of sensor measurements along orbit or biases from the algorithm itself, or both.

Although not perfectly symmetric, the aerosol index bias patterns show apparent positive biases for pixels where both solar zenith angle and satellite viewing angle are large. The western wing of an orbit appears to have more significant positive biases than eastern wing for both OMPS S-NPP and N20. The patterns are extremely similar between S-NPP and N20 with noticeable seasonal change from May to September. The highest positive errors for retrieved aerosol index have a scale of 1.2 at Northern hemisphere high solar zenith regions for the first 1-4 cross-tracks in September for both OMPS S-NPP and N20. Unlike the

positive biases, negative aerosol index biases are much milder over about 15 cross-tracks around nadir position for middle and high solar zenith angle regions, and those regions shift to western wing of the orbit when the solar zenith angle gets smaller. Those shift patterns are pretty similar for both OMPS S-NPP and N20 with different seasons. It is difficult to quantify how much of those biases are from measurement errors and how much is from the V8TOZ algorithm. Since the bias patterns are so close for both OMPS S-NPP and N20, suggesting a limited association with sensors, it is more likely that those error biases

are related to algorithms or errors in the SDR data. The potential errors from non-linearity and the inaccurate addressing of the relationship between aerosol and its effects on reflectivity in terms of various solar zenith angles and satellite viewing angles may play a role on retrieval biases.







**Figure 8.** The along orbit retrieved total column ozone biases in percent associated with 35 cross-tracks for NOAA OMPS S-NPP (upper
panel) and OMPS N20 (bottom panel). The left two plots represent the averaged ozone biases for the first and second half of May, 2020,
while the right two plots represent the averaged half-month ozone biases for September 2020.

### 4.1.2 Biases in retrieved TCO

The convention and statistical processing are the same as Fig. 7, Fig. 8 shows ozone retrieval biases in percent along the orbit
for 35 cross-tracks for both NOAA OMPS S-NPP and N20 products. Slightly different from aerosol bias patterns that are
nearly persistent between S-NPP and N20 for different seasons, the bias patterns for retrieved ozone are not so persistent, but
still very similar with respect to different sensors and different seasons. In aerosol retrievals, strong positive biases were
detected when both solar zenith angles and satellite viewing angles are large. However, for ozone retrievals, similar biases
occurred at the very edge of the regions with even higher solar zenith angles. Unlike aerosol biases, the signs of those retrieved
ozone biases are different with respect to different hemispheres. The significant positive biases at extreme high solar zenith
angle regions in the Northern hemisphere and apparently negative biases in the Southern hemisphere, may imply the ozone





biases for extreme conditions are complex. The measurement errors as well as non-linearity and inaccuracy of retrieval algorithm should have all contributed to those biases.

Besides those extreme regions with very high solar zenith angles for a few edged cross-tracks where the ozone biases could be more than ±2%, ozone biases for most areas are relatively mild. The bias patterns are slightly different for different sensors
at different hemispheres. The ozone biases also show somewhat seasonal change in magnitude which indicate the biases in September are slightly larger than May with a scale of less than ±1%. There is a tendency of mild positive ozone biases around middle 15 cross-tracks for both S-NPP and N20. Pixels with high satellite viewing angles seem to have slightly negative biases in retrieved ozone. Those bias patterns didn't show apparent features in symmetry, but in general, the bias structures are quite stable and persistent spatially and temporally. All those features indicate that the detected biases in ozone retrieval are not
random errors, they have to be related to biases either from measurements or from shortcomings in the algorithm. A proper correction would be able to minimize those systematic errors to a certain level.

## 5 Comparison with other products

Previous sections addressed the robustness and stability of retrieved total column ozone and aerosol index from NOAA OMPS S-NPP/N20 V8TOZ products. Similar products have been generated from various satellite instruments. In this section, we will
focus on comparing retrievals of NOAA OMPS V8TOZ with well-validated products from other satellites, such as Sentinel-5p TROPOMI and DISCOVR EPIC. The latter comparisons are somewhat circular as the EPIC was soft calibrated to agree with OMPS (See Geogdzhayev and Marshak 2018). We also compared retrievals with those from NASA processed OMPS S-NPP V8TOZ to further check the fidelity of our adjustments.

Due to differences in measuring time and method as well as algorithms used for retrievals, comparison of retrievals for
different products is usually based on zonal mean characteristics. In this study, we mainly focus on features of retrievals at grid level for more detailed comparison of the deviations associated with different satellites. To remove potential biases from measuring time, the grid values for all those products are generated based on pixels with scanning time differences roughly within two hours over equatorial areas. In the process of generating gridded total column ozone, pixels with very high SZA and SVA as well as data that do not have good quality are removed. From multi-sensor bias analysis, we notice that the
retrieved ozone biases that need multi-sensor correction are mainly present in the high SZA and high SVA regions. Therefore, there are no big differences between soft-calibrated TCO/AI and multi-sensor corrected TCO/AI after making grids. We used soft-calibrated TCO/AI retrievals of current operational OMPS V8TOZ at NOAA for the following comparison. Three days' retrievals from September 10 to September 12, 2020 were chosen for comparison, because at that period of time massive wildfires occurred on the West Coast of the United States which exhibit global scale impact on the environment.




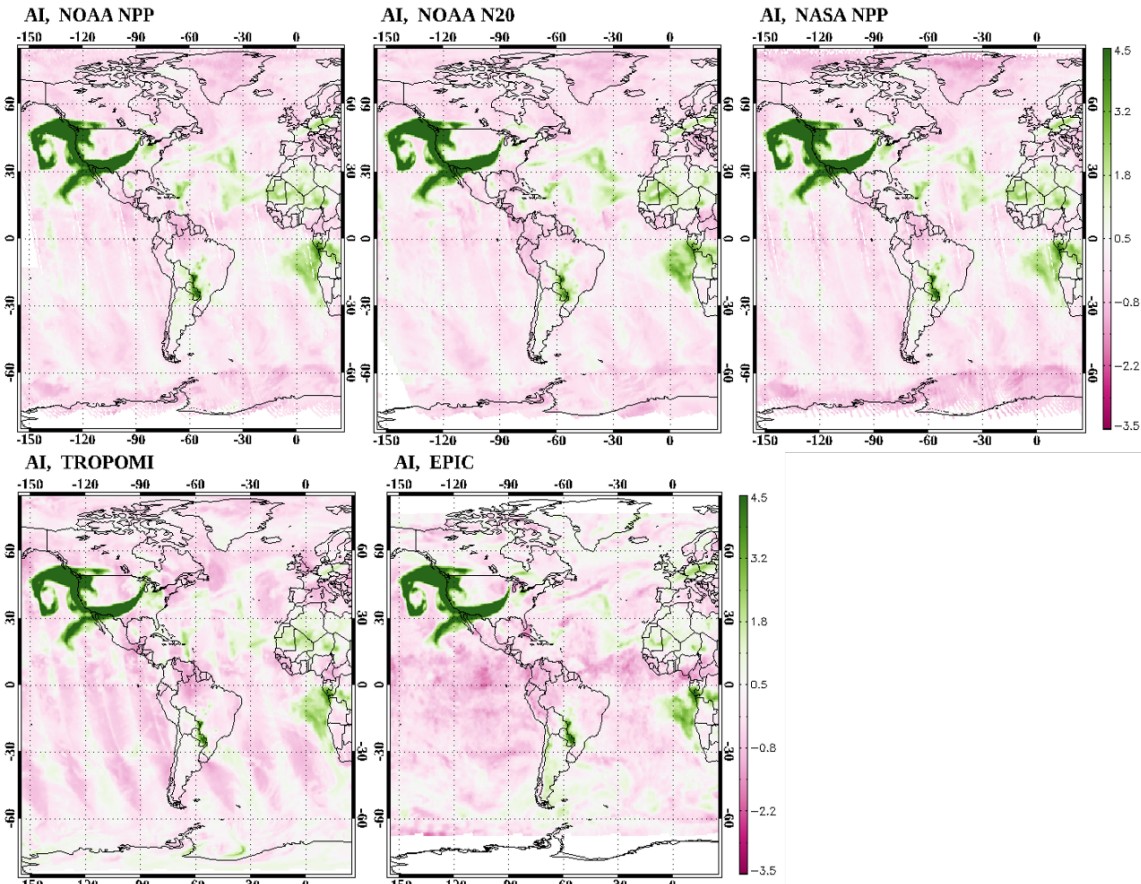

**Figure 9.** The gridded aerosol index for soft-calibrated NOAA OMPS S-NPP retrievals (upper left), soft-calibrated NOAA OMPS N20 retrievals (upper middle), NASA OMPS S-NPP retrievals (upper right), TROPOMI retrievals (bottom left) and EPIC retrievals (bottom right) for the date September 12, 2020. Those grid values were adjusted by their median values for scale consistency between products.

## 5.1 Differences in AI retrievals

Figure 9 shows gridded aerosol index (AI) retrievals from different satellites on September 12, 2020. As was defined in the previous section, the aerosol index is the difference between measured and calculated radiance with respect to a reference reflectivity spectral channel. AI products from different satellites differ to some degree in terms of differences in sensors and wavelengths used for deriving aerosol index. To make an explicit comparison, we re-adjust all those retrieved aerosol indexes in terms of median values from three days' (Sep.10 ~ Sep.12, 2020) global gridded AI data pool within 70N and 70S. The

median values (see Table 1) indicated that the mean states of retrieved aerosol index from different satellites or different processing vary significantly, with TROPOMI AI appearing to be much lower (-1.759) than any of the AI retrievals from other satellites. The median values of AI from OMPS and EPIC are pretty close to each other. Retrievals from EPIC are more likely to exhibit a relatively larger median AI value (0.454), and the AI value (0.262) from NASA processed OMPS S-NPP retrieval is likely to be about 0.13 lower than NOAA processed AI retrieval. All the products show clearly the massive wildfires in the

California, Oregon and Washington states of the USA, and the plumes associated with wind. The retrieved AI patterns





associated with this extreme wildfire appear to be very close to each other for different sensors in terms of regions and magnitude. Some small-scale wildfire events occurred in the central Southern America and west coast of central Africa are also detected from all those instruments. Compared to biomass loading from wildfire, AI from dust loading over the Sahara area is relatively weak for the date September 12, 2020. The magnitude of those dust related AI appear to be slightly smaller
from EPIC and TROPOMI retrievals than those from OMPS retrieval.

Except for those areas that show apparent mass loading signals, the close to real retrieval of aerosol index over other regions should be smooth with no features associated with geolocation and cloud patterns. However, for a general viewing of AI patterns in the Fig. 9, we saw that the base AI patterns from all the products still show noticeable cross-track related structures and biases associated with cloud patterns, which can be mostly removed in NOAA OMPS S-NPP and N20 by Multi-sensor
correction (figures not shown here). The base standard deviation statistics (see Table 1), which are calculated from those grids that have AI values less than median +1.2 for the three days within 70N and 70S, indicate that AI retrievals from OMPS V8TOZ have slightly less noise than those from TROPOMI and EPIC.

**Table 1.**   Statistics of AI for different sensors

| Sensors | NOAA NPP | NOAA N20 | NASA NPP | EPIC | TROPOMI |
|---|---|---|---|---|---|
| Median | 0.380 | 0.397 | 0.262 | 0.454 | -1.759 |
| Base-STDDEV | 0.407 | 0.400 | 0.424 | 0.569 | 0.510 |

**Table 2.**  Statistics AI for Scatter Density Plots in Fig. 10

| Two products | NOAA-N20 /NOAA-NPP | NASA-NPP /NOAA-NPP | EPIC /NOAA-NPP | TROPOMI /NOAA-NPP | TROPOMI /EPIC |
|---|---|---|---|---|---|
| R-square | 0.936 | 0.997 | 0.923 | 0.955 | 0.925 |
| Slope | 0.984 | 1.031 | 1.223 | 1.289 | 1.003 |
| Departure mean | -0.042 | -0.173 | -0.524 | -2.852 | -2.408 |
| Departure STDDEV | 1.000 | 0.228 | 1.371 | 1.219 | 0.925 |




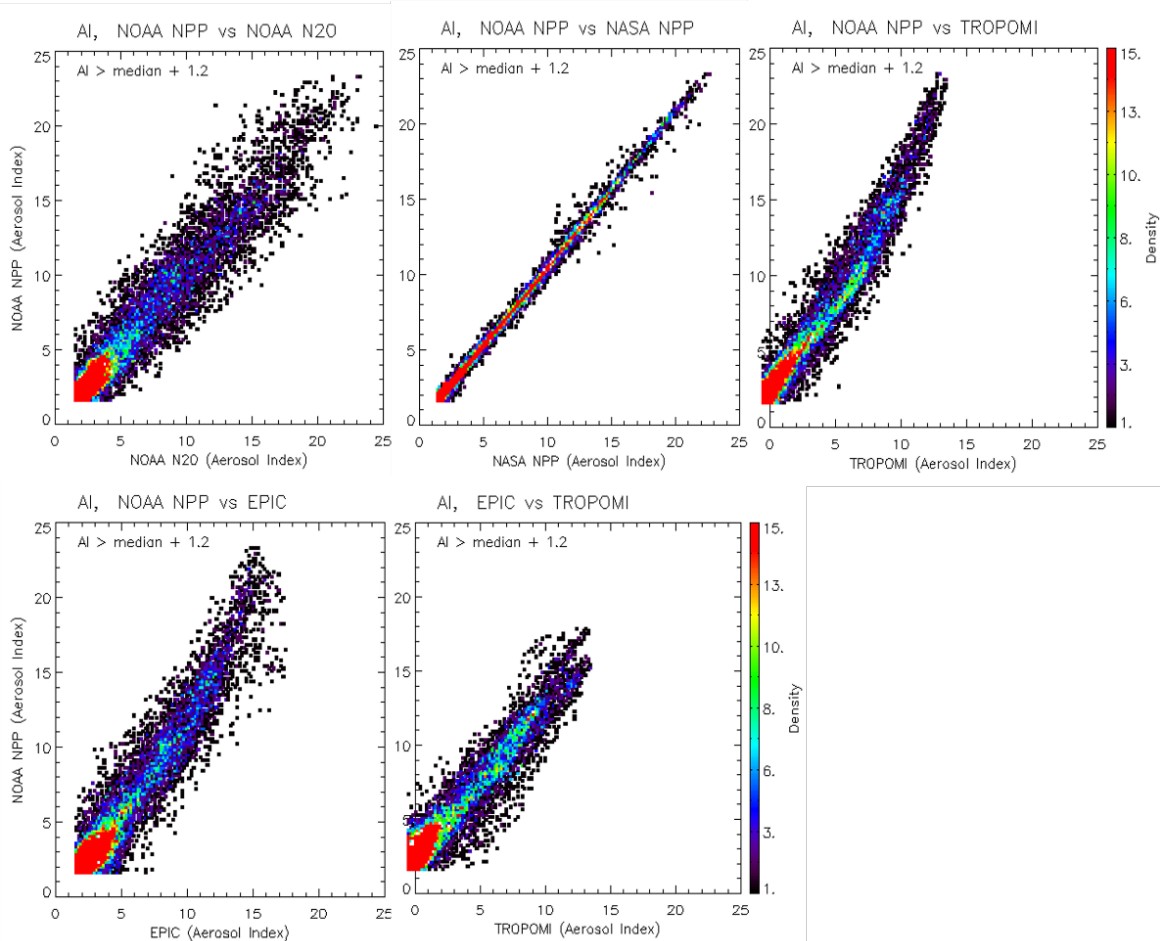

**Figure 10.** The scatter density plots of the gridded aerosol index between two products for NOAA OMPS S-NPP vs. NOAA OMPS N20 (upper left), NOAA OMPS S-NPP vs. NASA OMPS S-NPP (upper middle), NOAA OMPS S-NPP vs. TROPOMI (upper right), NOAA OMPS S-NPP vs. EPIC (bottom left) and EPIC vs. TROPOMI (bottom right) from three days' retrievals (Sept. 10~12, 2020). Those grids that AI > 1.2 were chosen for analysis and plotting.

Figure 10 illustrates the similarity of retrieved aerosol index from different products by scatter density plots. We mainly focused on those grids that both sensors exhibit discernible biomass or dust loading signals with AI > median + 1.2. Those plots would be able to show more explicitly the retrieval differences between two products in magnitude. As shown in Fig. 10, the retrieved AI events from NOAA OMPS S-NPP and NASA OMPS S-NPP appear to be very close, the r-square, slope, departure mean, which illustrates the difference in mean state, and the departure STDDEV, which shows the standard deviation of those selected grid values (see Table 2), are all indicate that the AI retrievals between those two products are almost the same. Those are expected results, and the slight differences come from some sort of differences in pixel geolocation, wavelength registration and soft-calibration adjustments between different research groups. The slope and departure mean between NOAA OMPS S-NPP and NOAA OMPS N20 (see Table 2) indicate that the retrieved AI from those two products



are very close in magnitude, the relatively large standard deviation with a relatively broader distribution in scatter density plot are mostly contributed by the movement of biomass or dust loading with wind due to the differences in measuring time.

While the retrievals of aerosol index from OMPS S-NPP and N20 exhibit strong linear relationship in magnitude with high correlation coefficient and slopes close to 1.0, the AI retrievals from both TROPOMI and EPIC, although showing high

correlation with OMPS retrievals (see Table 2) have a non-linear relationship with those from OMPS retrievals (see Fig. 10). To establish a proper exponential relationship of aerosol index, as shown in Fig. 10, between OMPS and other satellites is out of the scope of this study. We display this phenomenon here mainly since we want to demonstrate that there is potential scale and magnitude inconsistency in the AI products, which are associated with the wavelength pairs used for deriving aerosol index and the assumption in the models of the reflectivity dependence with wavelength. Those kinds of disagreement in the

retrieved aerosol index suggest that, if people want to compare biomass or dust loading events, or need to address spatial and temporal variation of AI patterns using various products, and develop conversion to apply to those AI values for accuracy and consistency.

## 5.2 Differences in TCO retrievals

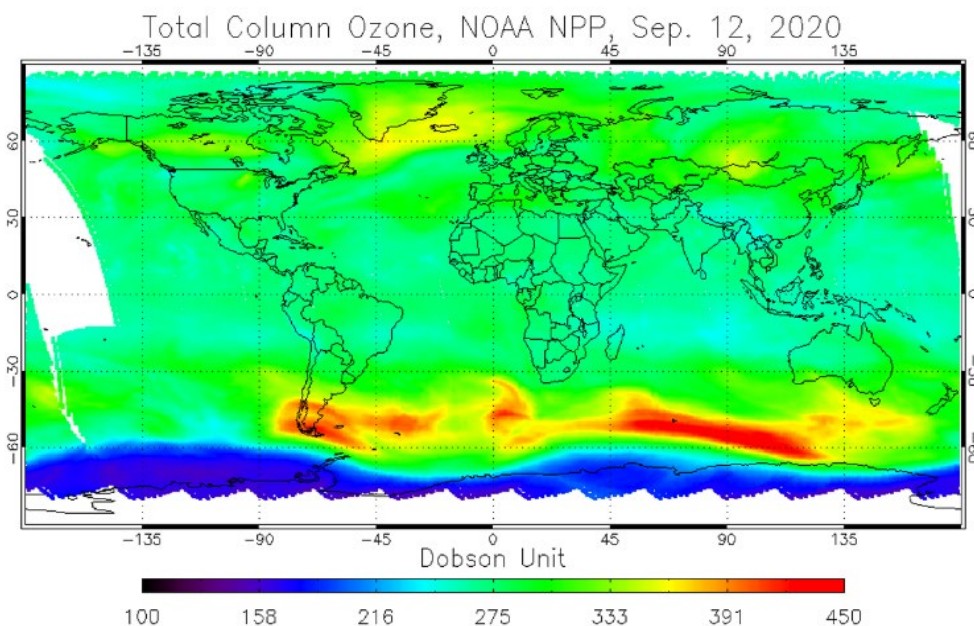

**Figure 11.** The map of total column ozone grids from soft-calibrated NOAA OMPS S-NPP retrievals for the date Sept. 12, 2020.

Figure 11 exhibits gridded total column ozone from NOAA OMPS S-NPP retrievals. It shows a typical global ozone pattern for September with a significant ozone hole over the Southern pole area surrounded by belt-like high ozone regions. Relatively lower and stable ozone over equatorial and middle latitude areas, while some weak high ozone centers appear over high latitude





regions in the Northern Hemisphere. The retrievals of total column ozone from other instruments were not shown here simply

because they are so similar to each other with little apparent differences from map viewing. Nevertheless, discernable

deviations in retrieved ozone associated with algorithms as well as different instruments can be seen by grids comparison and

scatter density plots from different products.

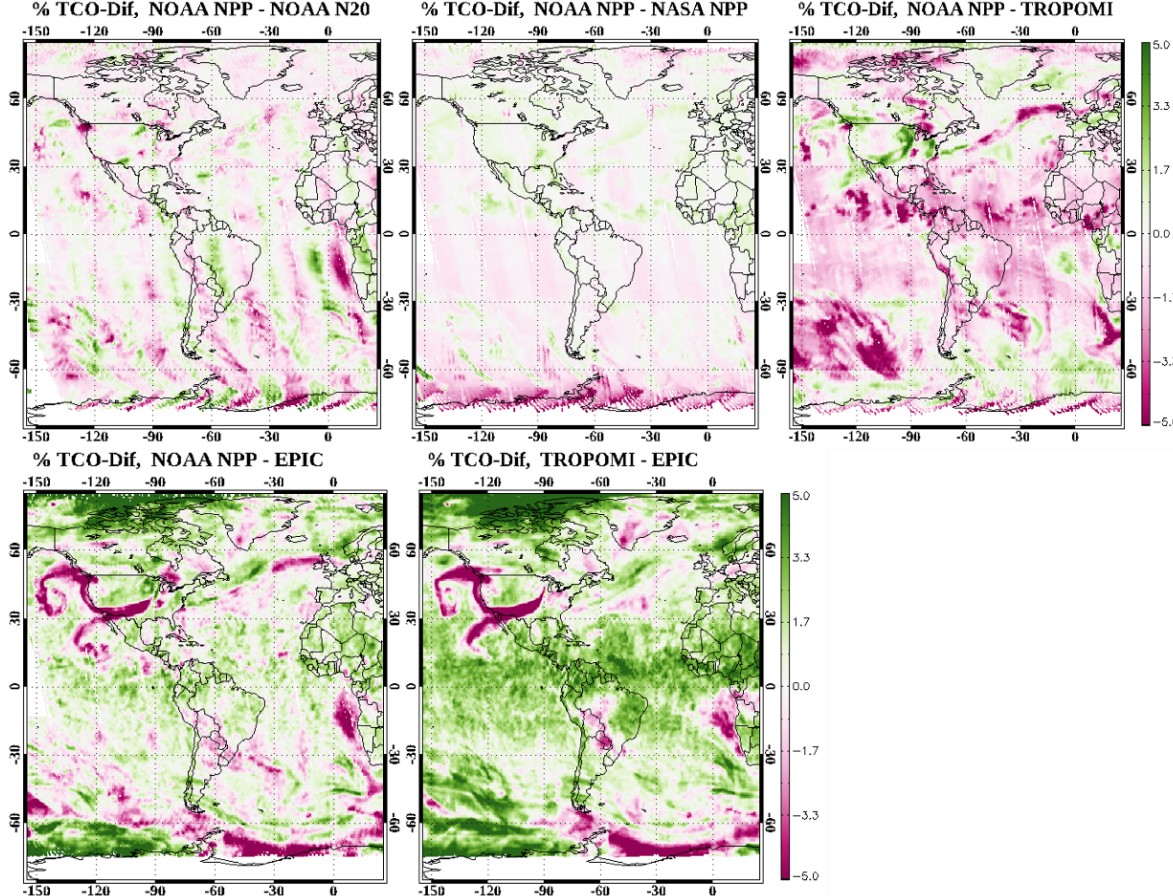

**Figure 12.** The map differences of retrieved total column ozone in percent between two products for NOAA OMPS S-NPP and NOAA
OMPS N20 (upper left), NOAA OMPS S-NPP and NASA OMPS S-NPP (upper middle), NOAA OMPS S-NPP and TROPOMI (upper
right), NOAA OMPS S-NPP and EPIC (bottom left) and between EPIC and TROPOMI (bottom right) for the date Sept. 12 2020.

Figure 12 illustrates the differences in percentage of retrieved total column ozone between two products in grid level for the

date September 12, 2020. Although there is a ~50 minutes' difference in measuring time, the differences between NOAA

OMPS S-NPP and N20 are pretty small. There are some noticeable cross-track related features along orbits. Those differences

in retrieved ozone are likely associated with the cross-track deviation pattern and the half-orbit shift between the two platforms.

The large differences of the coast of Africa are in a region of high aerosol loading. Some timing related deviation associated

with movement of smoke plumes, which cause some sort of noise in retrieved ozone, can be seen with weak signals. As

expected, the map comparison of retrieved ozone between NOAA OMPS S-NPP and NASA OMPS S-NPP shows that the




differences in the retrievals are usually less than +- 1%. The high agreement between those two products is also illustrated in the scatter density plot in Fig. 13, with close to 1 for both R-square and slope for the correlation of the ozone grids over 70N and 70S for the three days (Sep. 10 ~ Sep. 12, 2020) statistics in the Table 3. The averaged ozone retrievals for those two products are very close to each other with only 0.13 DU in difference. However, the departure STDDEV, which shows the standard deviation of the grid differences between NOAA OMPS S-NPP retrievals and NASA retrievals, indicates that those

two products still have discernable inconsistency with 1.67 DU standard deviation. By closely looking at the plot of map comparison, we would be able to see some kind of cross-track and cloud related features, which seems to be the main contributor to this standard deviation. As mentioned before, the slight differences in pixel geolocation, wavelength registration and soft-calibration adjustment might be the main reason for those deviations.

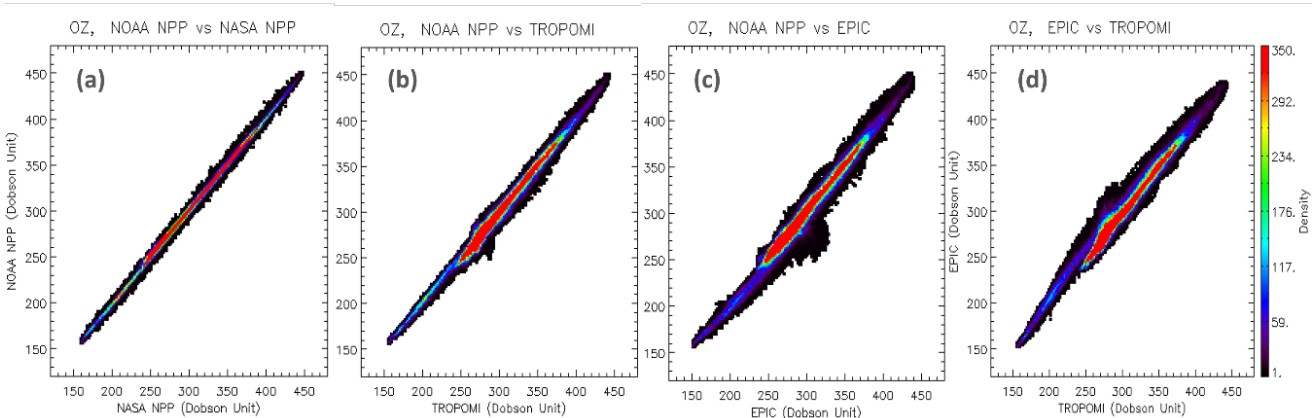

**Figure 13.** The scatter density plots of the gridded total column ozone between two products for NOAA OMPS S-NPP vs. NASA OMPS S-NPP (1st left), NOAA OMPS S-NPP vs. TROPOMI (2nd left), NOAA OMPS S-NPP vs. EPIC (3rd left) and EPIC vs. TROPOMI (4th left) from three days' retrievals (Sept. 10~12, 2020)

**Table 3.** Statistics of Ozone for Scatter Density Plots in Fig. 13

| Two products | NASA-NPP /NOAA-NPP | TROPOMI /NOAA-NPP | EPIC /NOAA-NPP | TROPOMI /EPIC |
|---|---|---|---|---|
| R-square | 0.998 | 0.989 | 0.980 | 0.976 |
| Slope | 1.004 | 1.017 | 1.002 | 1.000 |
| Departure mean | 0.130 | 1.940 | -1.360 | 3.316 |
| Departure STDDEV | 1.670 | 4.100 | 5.350 | 5.809 |

Unlike total column ozone retrievals from OMPS S-NPP and N20, which exhibit high level global similarity and consistency, retrievals from other satellites with different algorithms appear to show quite significant deviations compared to the retrievals from OMPS V8TOZ. The map comparison between TROPOMI and OMPS S-NPP (Fig. 12) exhibits apparent positive deviation of retrieved ozone for TROPOMI, with an averaged value of 1.94 DU from Table 3. However, the tendency of conducting slightly larger ozone retrievals from TROPOMI is not systematic, with about 3 to 4 percent positive deviation



occurring over cloud areas, and around 2 percent positive deviation over equatorial and middle latitude Southern hemisphere. In contrast to cloudy regions, ozone retrievals over mass loading regions appear to be slightly smaller than those from OMPS S-NPP, implying that there are differences in ozone corrections regarding impaction from clouds and aerosol index between those two algorithms. Scatter density plot (Fig. 13) of TROPOMI versus OMPS S-NPP shows no apparent discontinuity in ozone retrievals of those two products, with high correlation of 0.989 R-square and close to 1 slope (Table 3). Relatively large

standard deviation (4.1 DU) for those grid differences in ozone retrievals may partially come from the inconsistency in bias correction for clouds and aerosol index as mentioned above, as well as from differences in measuring time.

In contrast to comparison with TROPOMI, the map differences between EPIC and OMPS S-NPP (Fig. 12) shows apparent negative deviation of retrieved ozone for EPIC, with an averaged value of -1.36 DU from Table 3. Those negative deviations of EPIC retrievals appear to be pretty stable over equatorial areas with about 1 ~ 2 percent in most grids. However, the deviation

of ozone retrievals between EPIC and OMPS S-NPP exhibits quite inconsistent features over high latitude regions for both hemispheres. While the timing differences might be one of the reasons for those deviations, the scanning difference between those two satellites, which make big satellite viewing angles over high latitudes for EPIC measurements, could also contribute to those inconsistencies. One notable deviation between EPIC and OMPS S-NPP in retrieval is the apparent positive bias over significant aerosol loading regions, which indicates about 3 to 4 percent gain of retrieved total column ozone. This particular

bias suggests differences in correcting total column ozone retrieval by impact of aerosol loading between EPIC and S-NPP. Those deviations over aerosol loading areas and in the high latitude regions for both hemispheres appear to be responsible for the inconsistency in the scatter density plot in Fig. 13, with standard deviation for the differences between those two products reaching as high as 5.35 DU. In spite of those apparent inconsistencies, the high correlation of R-square equal to 0.98 with close to 1 slope in retrieved ozone between them still indicate that those are highly comparable products.

Since retrievals of total column ozone from EPIC appear to exhibit negative bias while retrievals from TROPOMI show positive bias compared to those from OMPS S-NPP, the large differences between EPIC and TROPOMI are expected as shown in Fig. 12. TROPOMI is likely to produce 3 to 4% larger ozone retrievals than those retrieved from EPIC over equatorial areas, this deviation getting smaller over middle latitude, but becoming less stable over high latitude regions. The opposite differences in retrievals also appear in the aerosol loading regions with EPIC being likely to generate as high as 5 percent larger ozone

retrievals than those from TROPOMI. Although the overall correlation indicates that EPIC and TROPOMI are quite comparable products in ozone retrievals with R-square equal to 0.976 and slope close to 1, those obvious discrepancy in retrievals could still be seen in the scatter density plot with big difference in mean (3.316 DU) and high standard deviation (5.809 DU) for the grid differences in ozone retrievals.



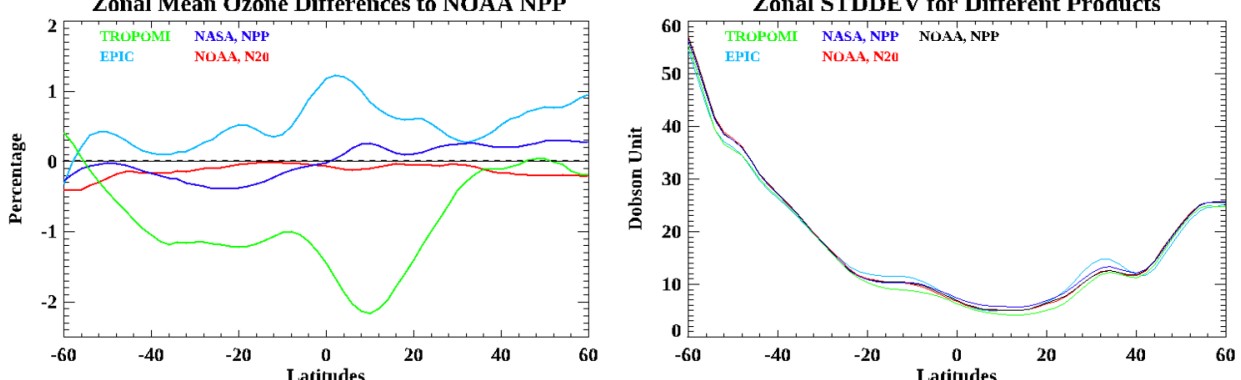

**Figure 14.** The left plot represents 10-degree moving zonal mean differences of from NOAA OMPS S-NPP with those from soft-calibrated NOAA OMPS N20 (red), from NASA OMPS S-NPP (dark blue), from TROPOMI (light blue), and from EPIC (green). The right plot has the same conventions but for the standard deviations of retrieved total column ozone.

In order to explore more details of the differences for the retrieved ozone between those products, we made two more plots in Fig. 14 based on three days' statistics, which show 10-degree zonal mean differences from 60S to 60N latitude for ozone retrievals as well as standard deviations of retrieved ozone compared to those from NOAA OMPS S-NPP. Since soft-calibration corrections conducted on the NOAA OMPS S-NPP and N20 aim at making retrievals be consistent, it is expected to see that ozone retrievals from NOAA OMPS products are in good agreement with very little discrepancy over global. It is interesting to see that NASA processed OMPS S-NPP retrievals are likely to be slightly larger than those from NOAA OMPS S-NPP over Southern hemisphere middle latitude regions, while showing opposite biased retrievals in the Northern hemisphere. In conducting soft-calibrations for NOAA OMPS V8TOZ, we forced monthly mean of AI and best guess of ozone to be flattened along cross-tracks. However, the averaged AI (Fig. 4-b) and best ozone (Fig.4-d) for NASA S-NPP V8TOZ still show some sort of positive and negative trends along cross-tracks respectively. Those differences in soft-calibrated retrievals and other sources of difference such as slight inconsistency in geolocation may explain the discrepancies between NOAA S-NPP and NASA S-NPP. Like map comparison, ozone retrievals from TROPOMI appear to produce larger ozone than those processed from NOAA OMPS, with around 1 percent high in the Southern hemisphere. There is up to 2 percent bias in retrieved ozone around 10N between TROPOMI and NOAA OMPS, which is more likely due to algorithm differences such as below cloud ozone corrections or adjustments for aerosols. In contrast to retrievals from TROPOMI, EPIC is likely to estimate smaller total ozone than those retrieved from NOAA OMPS, with about 0.5 percent low over global and around 1 percent low over equatorial region.

The variations of standard deviation of retrieved TCO along latitude for different products are pretty consistent, which show higher variability of ozone pattern over high latitude regions especially for Southern hemisphere, and the most stable ozone exists over the region from 0N to 20N. Those are typical ozone variability patterns for September. It is expected, if a similar algorithm is applied in ozone retrieval for different instruments, similar magnitude of retrieved ozone will be estimated, with less variability representing more homogeneous retrievals. We saw that in general NASA retrieved ozone appear to have slightly more variability than those from NOAA processed retrievals. It is likely using broadband spectrum for ozone and





reflectivity channels would reduce noise for the total column ozone retrieval. There are some differences in zonal mean standard deviation among NOAA OMPS, TROPOMI and EPIC. Since they used different algorithms in retrieval, it is difficult to tell if the differences in the variability come from the algorithm or from level-1 solar data.

## 6 Summary and Conclusions

The V8TOZ algorithm with narrowband spectra has been employed for NOAA satellites in total column ozone and aerosol index retrievals for previous years, a switch to using broadband spectra for the ozone and reflectivity channels has been implemented at NOAA operational system for conducting ozone and AI retrievals. This study mainly focused on addressing the stability and improvement when using broadband approach; establishing soft-calibration adjustments for both OMPS S-NPP and N20; analyzing error biases; and comparing total column ozone and aerosol index retrievals from NOAA OMPS with products from other satellite instruments.

An apparent advantage of using broadband channels in retrievals is that it improves signal-to-noise ratios, and reduces sensitivity to Ring effects, stray light and wavelength scale shifts. The comparison of along orbit homogeneity deviation for the retrieved total column ozone indicates that retrievals with broadband approach appear to conduct more stable and consistent retrievals for both OMPS S-NPP and N20, and those improvements seem to be more apparent at low slant column density regions. It is likely that using broader bandpasses would be able to reduce retrieval biases as well as making comparable products from different sensors.

N-value sensitivities to the change of total column ozone and reflectivity were used to build soft-calibration adjustments for both OMPS S-NPP and N20. To ensure the calculated N-value adjustments are consistent for global application, two particular regions were chosen for deriving the soft-calibration. 1) The equatorial Pacific region was chosen for making reflectivity adjustment. We forced the averaged one-percentile reflectivity over that area for both OMPS S-NPP and N20 to be the same as those from NASA OMPS S-NPP retrievals. Cross-track related features, such as sun-glint hump and viewing angle effects from haze, aerosol and fair-weather cumulus clouds, have been carefully considered in correcting reflectivity at various cross-track positions. 2) The land areas between 25°S and 25°N were chosen for generating soft-calibration parameters for the other channels. We forced the averaged total column ozone, aerosol index as well as initial residuals of shorter wavelengths to be the same as those from NASA OMPS S-NPP, and made them flatten along 35 cross-tracks for the month March 2020. Independent verification experiments with V8TOZ retrievals from both OMPS S-NPP and N20 for the month September 2020, indicate that the soft-calibrations we built based on the described process are robust, and capable of providing stable and consistent retrievals for total column ozone and aerosol index.

Although the soft-calibration adjustments have forced the retrievals to be in agreement for the mean state between OMPS S-NPP and N20, we found there are still apparent cross-track related biases along orbit, especially over high latitude regions for both total column ozone and aerosol index retrievals. Those biases appear to exhibit strong association with both solar zenith angle and satellite viewing angle with minor seasonal change for both OMPS S-NPP and N20. Because intra-cross-track biases



and scale differences for both OMPS S-NPP and N20 have been mostly removed by soft-calibrations over equatorial regions,
those remaining biases are likely associated with either inconsistent biases of sensor measurements along an orbit or biases
from the algorithm itself. For instance, the true ozone profile shape differences from the standard profiles interacting with the
layer ozone retrieval efficiency factors will lead to small but complex cross-track retrieval errors. The Aerosol Index will be
affected by errors in the simple wavelength-independent effective reflectivity model as a function of SZA and SVA, leading to
cross-track biases, which further lead to ozone errors in Step 3 ozone, especially for retrievals with higher SZA and SVA.
Those are systematic like errors existing in both OMPS S-NPP and N20 retrievals, which could be removed or at least alleviated
by applying further bias corrections in the products.

    We also conducted a detailed comparison of NOAA OMPS retrievals based on the broadband approach with other well
calibrated products. We found that the retrieval algorithms, channel wavelengths used for deriving retrievals and differences
in measuring time could potentially contribute to the deviations in the products between different satellites. In general,
TROPOMI derived total column ozone values appear to be slightly larger than those retrievals from OMPS V8TOZ, while
EPIC is likely to generate somewhat lower ozone retrieval than OMPS. For the retrievals of aerosol index, TROPOMI appears
to have a scale shift in magnitude compared to that from OMPS and EPIC and both EPIC and TROPOMI AI estimates have a
nonlinear relationship with OMPS AI traceable to the specific wavelength pairs used in each product. Nevertheless, the overall
retrievals between those products are quite similar and consistent, even though the retrievals we chose for comparison were
impacted by extreme wildfires.

**Disclaimer and Acknowledgement**

This Article contents are solely the opinions of the authors and do not constitute a statement of policy, decision, or position on
behalf of NOAA or the U.S. Government. This work was supported by NOAA JPSS Program.

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
