# Peer review of "An Approach to Track Instrument Calibration and Produce Consistent Products with the Version-8 Total Column Ozone Algorithm (V8TOZ)"

_EGUsphere, 2022_

## Referee Comment (RC1)

**Review**
**Title**: An Approach to Track Instrument Calibration and Produce Consistent Products with the Version-8 Total Column Ozone Algorithm (V8TOZ)
**Authors**: Zhihua Zhang, Jianguo Niu, Lawrence E. Flynn, Eric Beach, and Trevor Beck

This paper describes an approach for radiometric adjustments of UV channels (between 310 – 380 nm) to achieve product consistency among viewing angles, demonstrate that the broad-band retrieval improves product quality over that of narrow-band retrieval. Papers that describe soft-calibration method are rarely published or submitted, but much needed. The broad-vs-narrow band finding is significant. Hence, I recommend publication of this paper, after addressing items listed below.

General comments
    This soft-calibration method improves the radiometric calibration of the OMPS instruments on SNPP and NOAA-20 satellites. However, this approach uses a soft-calibrated product (i.e., NASA's NMTO3-L2) as a reference. Therefore, its success depends on the success of a different soft-calibration method. Why develop a new one and not adopt the NASA method? How to ensure consistency over time from years to decades?

Specific comments
  1. Line 55: 'homogeneous' is not the right word to describe SDR.

  2. Line 74: replace 'statistical' with algorithmic.

  3. Line 74-75: 'The science basis and statistical procedures as well as error sources for the V8 algorithm have been well documented in the OMPS ATBD and other articles (Bhartia & Wellemeyer, 2002; McPeters et al., 1996).'

      Many $O_3$ errors depend on instrument characteristics, therefore error analyses need to be performed for each instrument (or each slit function).

  4. Line 78: 'measurement departure', not clear in the context.

  5. Line 79: 'Researchers interested in error analysis and refined retrievals could take it as reference.' Explain.

  6. Line 88-90: 'The first assumption is that the BUV radiances at wavelengths greater than 310 nm are primarily a function of total ozone amount, with only a weak dependence on ozone profile shapes that can be accounted for by using a set of climatological profiles.'

      This is NOT a good assumption for high (viewing and/or solar) zenith angles.

  7. Line 90-92: 'The second assumption is that a relatively simple radiative transfer model that treats clouds, aerosols, and surfaces as Lambertian reflectors can account for most of the spectral dependence of BUV radiation.'

Lambertian representation of surface and atmospheric particles (i.e., clouds and/or aerosols) works because radiative transfer through this simplified model atmosphere-surface system closely simulate those in the actual atmosphere, especially in the stratosphere, where most $O_3$ absorption happens (see Huang and Yang, doi: 10.5194/amt-15-5877-2022).

8. 'Account for most of the spectral dependence of BUV radiation' is a manifestation of the success of this simplified model, not an assumption.

9. Lines 129 – 130: 'The slit functions provide key information for the spectra convolved values of the ozone absorption cross-sections'.

   This statement seems to imply an incorrect construction or usage of look-up tables (LUTs). The correct LUT approach: 1) high-spectral resolution LUTs are constructed from radiative transfer calculations, 2) solar-weighted slit convolution of terms of Eq. 1 to create instrument (slit-function) specific LUTs. In these steps, slit-convolved cross-sections are not used.

10. Line 250: 'should keep the same value for 35 cross-track positions.' This description is not clear. Need revision.

11. Section 5, Comparison with other products

    There is another EPIC total $O_3$ product, which provides high-accuracy $O_3$ retrievals (based on the publication, Huang and Yang, doi:10.5194/amt-15-5877-2022). It is expected to have a higher correlation and lower spread between this EPIC product and the OMPS products from SNPP and NOAA-20. Please include this product in the inter-comparisons.

---

## Author Comment (AC1)

**Reply to Comments from Anonymous Referees**

**Comments from Anonymous Referee #1**

**Review**
**Title**: An Approach to Track Instrument Calibration and Produce Consistent Products with the Version-8 Total Column Ozone Algorithm (V8TOZ)
**Authors**: Zhihua Zhang, Jianguo Niu, Lawrence E. Flynn, Eric Beach, and Trevor Beck

This paper describes an approach for radiometric adjustments of UV channels (between 310 – 380 nm) to achieve product consistency among viewing angles, demonstrate that the broad-band retrieval improves product quality over that of narrow-band retrieval. Papers that describe soft-calibration method are rarely published or submitted, but much needed. The broad-vs-narrow band finding is significant. Hence, I recommend publication of this paper, after addressing items listed below.

General comments
> This soft-calibration method improves the radiometric calibration of the OMPS instruments on SNPP and NOAA-20 satellites. However, this approach uses a soft-calibrated product (i.e., NASA's NMTO3-L2) as a reference. Therefore, its success depends on the success of a different soft-calibration method. Why develop a new one and not adopt the NASA method? How to ensure consistency over time from years to decades?

The NASA S-NPP OMPS total ozone product used soft calibration from ice radiances for its reflectivity channel calibration but it also used comparisons to the NOAA-19 SBUV/2 ozone amounts for its ozone channel calibration. We want to tie the OMPS ozone record to the SBUV/2 record. Since the NOAA and NASA products for S-NPP use the same measurements, it is straightforward to make the two agree. The convergence of the products (NASA's and NOAA's) lessens confusion from multiple versions. While it is important to have good calibration for all of the channels used in the V8TOz, small errors in the absolute calibration of the 331 nm reflectivity channel will be partially mitigated by the development of adjustments to the 318 nm channel to match "truth" ozone values using those biased reflectivity results.
While the S-NPP products' calibration for the reflectivity channels thus trace their values back to the ice radiances, the cross-track dependence of the effective reflectivity and aerosol index over open ocean and vegetative land can be studied to check the performance of those adjustments. As discussed in the paper, the NOAA-20 cross-track reflectivity dependence was preserved from the laboratory calibration with an adjustment of the average level. The result, along with the pattern for minimum reflectivity over land, suggests that the ice radiance results may not be as good at higher view angles. Further, the long-term stability of the 1-percentile reflectivity over the Pacific box region is validated by the results in Figure 6.a taken with the instrument degradation shown in Figure 1. Good features of the Pacific box are that comparisons can be made all year round, that the solar zenith angles are low, and that the ozone is relatively stable and homogenous at the 10% level. This allows good cross-calibration of multiple sensors just using statistical matchups over multi-day coincident measurements.

Specific comments
1. Line 55: 'homogeneous' is not the right word to describe SDR.
   Agree.
   *The S-NPP OMPS-NM was reprocessed with a consistent set of calibration tables to produce an SDR data set of uniform quality (Zou et al., 2020; Yan et al., 2022).*

2. Line 74: replace 'statistical' with algorithmic.

*Agree*
*The science basis and algorithmic procedures as well as error sources for the V8*
*algorithm have been well documented ...*

3. Line 74-75: 'The science basis and statistical procedures as well as error sources for the V8 algorithm have been well documented in the OMPS ATBD and other articles (Bhartia & Wellemeyer, 2002; McPeters et al., 1996).'

   Many $O_3$ errors depend on instrument characteristics, therefore error analyses need to be performed for each instrument (or each slit function).

   As the reviewer knows, the instrument radiative transfer look-up tables are designed to account for variations in the slit functions. Further the OMPS NM instrument designs and SDR processings are very similar, and we have used the broad channel approach to lessen certain error sources impacts.

4. Line 78: 'measurement departure', not clear in the context.
   Changed:
   *Thanks to the OMPS series, which provide similar instruments with the same scanning method and the same local Equator crossing times in the same orbital plane,*

5. Line 79: 'Researchers interested in error analysis and refined retrievals could take it as reference.' Explain.
   Reworded.
   *Researchers interested in detailed error analysis and refined retrievals can take it as starting point. For example, investigate the residual errors present from the OMPS NM polarization sensitivity.*

6. Line 88-90: 'The first assumption is that the BUV radiances at wavelengths greater than 310 nm are primarily a function of total ozone amount, with only a weak dependence on ozone profile shapes that can be accounted for by using a set of climatological profiles.'

   This is NOT a good assumption for high (viewing and/or solar) zenith angles.

Agreed, caveat added.
*The first assumption is that the BUV radiances at wavelengths greater than 310 nm are primarily a function of total ozone amount, with only a weak dependence on ozone profile shapes that can be accounted for by using a set of climatological profiles. This is not a good assumption when the optical path length becomes large, e.g., at high solar zenith angles for large ozone loading.*

7. Line 90-92: 'The second assumption is that a relatively simple radiative transfer model that treats clouds, aerosols, and surfaces as Lambertian reflectors can account for most of the spectral dependence of BUV radiation.'

Lambertian representation of surface and atmospheric particles (i.e., clouds and/or aerosols) works because radiative transfer through this simplified model atmosphere-surface system closely simulate those in the actual atmosphere, especially in the stratosphere, where most $O_3$ absorption happens (see Huang and Yang, doi: 10.5194/amt- 15-5877-2022).

Agreed. We have added the reference.
*(See Huang and Yang, doi: 10.5194/amt-15-5877-2022.)*

8. 'Account for most of the spectral dependence of BUV radiation' is a manifestation of the success of this simplified model, not an assumption.

The algorithm developers were indeed smart guys, however we think that the use of "assumed" is acceptable here. Maybe "intuited", "recognized" or "expected", might be better.
We do not consider this paper to be a good place to have an expanded discussion of the errors present in the partial cloud model from differences in the computed cloud fraction versus the actual geometric cloud fraction, the cloud pressure in the model versus the true cloud top optical centroid, the cloud reflectivity versus the 80% model assumption, or the surface reflectivity wavelength dependence versus the actual dependence. An analysis of the performance shows that most are effects are second order, that is, they are usually products of two small errors.

9. Lines 129 – 130: 'The slit functions provide key information for the spectra convolved values of the ozone absorption cross-sections'.

This statement seems to imply an incorrect construction or usage of look-up tables (LUTs). The correct LUT approach: 1) high-spectral resolution LUTs are constructed from radiative transfer calculations, 2) solar-weighted slit convolution of terms of Eq. 1 to create instrument (slit-function) specific LUTs. In these steps, slit-convolved cross-sections are not used.

Yes, this was too simplistic a statement. Revised
*The slit functions provide key information for the spectra convolved values of the ozone absorption cross-sections as computed through the instrument table formulation using weighted averages of monochromatic radiance and irradiance components.*

10. Line 250: 'should keep the same value for 35 cross-track positions.' This description is not clear. Need revision.

Yes, this is not clear. We added description at line 250:
*In this study, the adjustments for the other channels were set to produce constant measurement residuals with no cross-track variation. The mean residuals for the channels were set at the target retrievals from NASA OMPS S-NPP V8TOz using comparisons over the equatorial land areas with cloud-free pixels. There are no sun-glint bumps to influence the residuals along the 35 cross-tracks.*

11. Section 5, Comparison with other products

There is another EPIC total $O_3$ product, which provides high-accuracy $O_3$ retrievals (based on the publication, Huang and Yang, doi:10.5194/amt-15-5877-2022). It is expected to have a higher correlation and lower spread between this EPIC product and the OMPS products from SNPP and NOAA-20. Please include this product in the inter-comparisons.

Thanks for the information. We will use this product in future work.

---

## Author Comment (AC2)

**Reply to Comments from Anonymous Referees**

**Comments from Anonymous Referee #2**

**General comments**

I enjoyed reading this paper, as harmonization of datasets is an important aspect in intercomparison of measurements and the interpretation of trends in timeseries, for example related to climate aspects. More specific, the paper focuses on a softcalibration technique to harmonize the total ozone measurements of the S-NPP and N20 OMPS sensors.

Overall, the paper is well written, although also some questions arose while reading, as indicated below.

I agree to publication of this paper, if the authors attend to the specifics below.

**Specific comments:**

**Throughout the manuscript**

I have the impression that the included imagery suffers from insufficient resolution, something that should be easily remedied.

  All figures were updated with higher resolution.

**1 Introduction**

Overall a clearly written introduction, providing sufficient justification and background for the use of the V8TOZ algorithm and sufficient references to relevant literature. I would like to read a little more here, though, on the motivation of the study. Why is the development of a new soft-calibration scheme required? What is missing in existing schemes (NASA) or why can't those be applied? Little by little the answers are given elsewhere in the paper, but the motivation should already be clear here.

  We added the following in the middle of Line 75 as a new paragraph:
*The results in this paper use soft calibration adjustments to force agreement between V8TOz retrievals for S-NPP and NOAA-20 with plans to continue using the method for NOAA-21 OMPS and other instruments. The adjustment method uses statistical comparisons over a latitude / longitude box over the equatorial Pacific. This region is selected for a variety of reasons including the following: 1) The total column ozone amounts are modest and the solar zenith angles are low; 2) The ozone profiles are relatively stable and consistent over the region (with some intra- and inter-annual and quasi-biennial*

*changes); 3) The atmospheric aerosol and SO$_2$ loading are usually close to background levels; and 4) The ocean surface presents a target with little intra-annual variability.*

**2. V8TOZ with a broader bandpass approach**

The broadband approach is described convincingly and is later shown to reduce retrieval noise and product biases. In section 2.4, the broadband approach is tested on one month of V8TOZ runs. Any reason to specifically choose this month and year? Same question for data selection further on in the paper.

   No, the range of ozone, reflectivity and aerosol loading over any month for the whole globe would have served as a good test of the wider channel method. The broadband approach has been implemented in the NOAA operations since July, 2022.

Line 204: see Fig. 3, left panel --> see Fig. 3, right panel.

   Agreed. Corrected.

**3. Soft-calibration for both OMPS S-NPP and N20.**

In Section 3.1, it becomes clear that OMPS S-NPP V8TOZ retrieval results from NASA are used as reference data set. At the same time, it appears that the existing soft-calibration method developed at NASA cannot be applied to the NOAA datasets, because of different treatment of the measurement data. This should be made clear earlier in the paper.

   The NASA soft calibration using ice radiances could have been applied. That calibration is primarily for the reflectivity channels and requires seasonal observations. The equatorial Pacific is available year round with good viewing conditions. Both methods use the cross-track reflectivity over dark vegetative scenes as a check on the performance. The soft calibration for ozone for the NASA S-NPP used comparisons to the ozone amounts from the NOAA-19 SBUV/2 retrievals. The equatorial Pacific presents a low variability ozone field for inter-instrument result comparisons to estimate ozone channel adjustments to force agreement.

Line 287: "The reflectivity from **the** broadband approach is generally **slightly** lower with less variation than **the** narrowband approach."

   Corrected.

Line 294: agree --> agreement

   Corrected.

Line 417: 0.3 --> 0.3%.

   Corrected.

**4 Errors and uncertainties versus latitude.**

In chapter 4, the authors show extensive tests and comparisons. I appreciate that they don't shy away from mentioning remaining uncertainties or lack of explanation of the origin of observed remaining biases.

    Thanks.

**5 Comparison with other products**

In the comparison of OMPS aerosol index and ozone column with those from other satellite sensors, it appears that ozone columns from NOAA OMPS show differences when compared to TROPOMI and EPIC. although suggestions are given to explain these differences, some uncertainty remains in these explanations (different factors contribute) and comparison with ground measurements for specific scenes would smaybe be useful. I do not ask to ask the authors to add a full section on comparison with ground data, but literature may also provide some sources that may help explain the observed offsets. After all, NASA OMPS S-Npp data is taken as reference. is this data known to agree better with validation data than TROPOMI and EPIC?

    For the Aerosol Indices, we plan to conduct a simple study creating retrievals using a range of short and long wavelength channel pairs with V8TOz partial cloud approach to quantify the effects of channel choices on those values and establish their functional relationship. We agree that the results in Figure 12 show a need for additional validation studies and analysis for the ozone estimates.

---

## Author Response (AR2)

**Reply to Comments from Editor Mark Weber**

**Review**
**Title**: An Approach to Track Instrument Calibration and Produce Consistent Products with the Version-8 Total Column Ozone Algorithm (V8TOZ)
**Authors**: Zhihua Zhang, Jianguo Niu, Lawrence E. Flynn, Eric Beach, and Trevor Beck

**Public justification (visible to the public if the article is accepted and published)**:
Dear Zhihua Zhang,

your paper is nearly ready to be published in AMT. There is one minor issue which needs to be addressed. Both reviewers remarked that little motivation in the introduction is given why a new of soft calibration for OMPS is needed in the light of an existing soft-calibration approach applied to SBUVs using ice radiances. The changes you made do not completely answer this. In your reply to the reviewers, you explain that ice radiances are insufficient. So please summarise in the introduction briefly the ice radiance calibration (incl. references) and explain why it is insufficient for OMPS-NM before you explain the new approach as done in your revision.

Best wishes,
Mark Weber

Dear Mark Weber,

A new paragraph as below starting on line 83 to line 96 was added to the Introduction that gives why a new of soft calibration for OMPS is needed in the light of an existing soft-calibration approach applied to SBUVs using ice radiances:

We start the process with the NASA S-NPP V8TOZ products. The Level 1 data records for those retrievals were calibrated to give agreement with NOAA-19 SBUV/2 total ozone at the start of the record and cross-track adjustments from ice radiance studies were used to set the effective reflectivity (McPeters et al., 2019). The NASA soft calibration using ice radiances could have been applied here for NOAA-20. That calibration is primarily for the reflectivity channels and requires seasonal observations. The equatorial Pacific is available year round with good viewing conditions and better stability. Further, the close to identical Equator-crossing-times of S-NPP, NOAA-20 and NOAA-21 allow direct comparisons of the cross-track reflectivity, aerosol and ozone patterns over that region. Both methods use the cross-track reflectivity over dark vegetative scenes as a check on the performance. The soft calibration for ozone for the NASA S-NPP used comparisons to the ozone amounts from the NOAA-19 SBUV/2 retrievals. This data set is not available for the NOAA-20 OMPS NM. Fortunately, the NPP OMPS NM dual diffuser system has been working well to track the small levels of instrument degradation. This means that ozone comparisons between NPP OMPS NM and NOAA-20 OMPS NM give a good approach for generating a consistent addition to extend the long-term record. The equatorial Pacific presents a low variability ozone field for inter-instrument result comparisons to estimate ozone channel adjustments to force agreement. The V8TOZ Radiative Transfer Lookup tables and retrieval algorithm act as a transfer between the two OMPS NM measurements at the 12 channels used in the V8TOZ.